# ANO : FASTER IS BETTER IN NOISY LANDSCAPES

## ABSTRACT

Stochastic optimizers are central to deep learning, yet widely used methods such as Adam and Adan can degrade in non-stationary or noisy environments, partly due to their reliance on momentum-based magnitude estimates. We introduce Ano, a novel optimizer that decouples direction and magnitude: momentum is used for directional smoothing, while instantaneous gradient magnitudes determine step size. This design improves robustness to gradient noise while retaining the simplicity and efficiency of first-order methods. We further propose Anolog, which removes sensitivity to the momentum coefficient by expanding its window over time via a logarithmic schedule. We establish non-convex convergence guarantees with a convergence rate similar to other sign-based methods, and empirically show that Ano provides substantial gains in noisy and non-stationary regimes such as reinforcement learning, while remaining competitive on low-noise tasks.

## 1 INTRODUCTION

Stochastic optimization is central to modern deep learning. Adaptive methods such as Adam (Kingma & Ba, 2015), and their variants (Reddi et al., 2018; Loshchilov & Hutter, 2019; Zaheer et al., 2018) are widely used because they automatically adjust step sizes and often accelerate early training. However, their behavior can degrade under noisy or non-stationary conditions: mini-batch stochasticity and data augmentation induce gradient noise (Mandt et al., 2017), labels may be ambiguous or noisy (Zhang et al., 2017; Song et al., 2022), and in reinforcement learning the training targets evolve over time (Henderson et al., 2018; Mnih et al., 2016). A key limitation is that Adam couples update direction and magnitude through momentum: prior work (Balles & Hennig, 2018) shows that the momentum sign already captures most directional information, while its magnitude and the second-moment estimate impose heavy smoothing.

We propose **A New Optimizer**, abbreviated as **Ano**, designed to handle noisy optimization landscapes. First, we decouple direction and magnitude: updates follow the momentum sign for stability, while the step size is scaled by an SNR-like ratio of instantaneous gradients, avoiding the sluggishness of momentum-based magnitudes. Second, we revisit Yogi's asymmetric variance update (Zaheer et al., 2018), which accelerates recovery after noise spikes, and introduce an additional decay factor to control its memory. This preserves Yogi's fast responsiveness while ensuring smoother adaptation under highly stochastic gradients.

We summarize our main contributions as follows:

- We propose a new gradient-scaling mechanism that removes the reliance on momentum-based magnitude estimates, leading to better adaptation in non-stationary and/or noisy optimization landscapes with the same memory cost as Adam.

- We provide a theoretical analysis of Ano in the non-convex setting, establishing a convergence rate of $\mathcal{O}\left(k^{-1/4}\right)$ under standard assumptions, matching existing results for sign-based optimizers.

- We evaluate Ano on supervised and reinforcement learning tasks, showing clear gains in noisy, non-stationary settings while remaining competitive in standard benchmarks.

## 2 RELATED WORK

Research on stochastic optimizers spans several directions. We briefly review the lines most relevant to Ano and situate our contribution.

**Adaptive methods.** AdaGrad (Duchi et al., 2011), AdaDelta (Zeiler, 2012) and RMSProp (Tieleman & Hinton, 2012) pioneered coordinate-wise adaptivity; Adam (Kingma & Ba, 2015) combined first- and second-moment estimates and became a default in deep learning. Yogi (Zaheer et al., 2018) stabilizes the second-moment accumulator for non-stationary regimes. More recently, Adan (Xie et al., 2024) couples adaptive moments with Nesterov-style lookahead and has emerged as a competitive baseline. Our optimizer Ano relates to this family through variance-aware step-size control, but differs in how direction and magnitude are constructed.

**Sign-based methods.** SignSGD and Signum (Bernstein et al., 2018) reduce updates to element-wise signs, offering scale invariance and communication efficiency in distributed settings. Lion (Chen et al., 2023) revisits sign-based updates with a tailored momentum schedule, yielding strong empirical results. Ano keeps the robustness of sign-informed directions but reintroduces gradient magnitudes through an explicit decoupling, trading pure scale invariance for finer adaptivity.

**Direction–magnitude decoupling.** Recent works such as Grams (Cao et al., 2024) decouple the update by using gradient signs for direction and the momentum norm for scaling. Ano adopts a complementary design: momentum provides a stable directional signal, while the raw gradient norm sets the step size. This hybridization aims to combine the resilience of sign-based directions with the adaptivity of moment estimators.

**Optimization under non-stationarity.** Non-stationarity is a known stressor for optimization, particularly in RL. Prior work tackled it with task-specific procedures such as *Normalize and Project (NaP)* (Lyle et al., 2024) and meta-learned optimizers for RL (Lan et al., 2025). While these highlight the need for robustness to evolving objectives, they are not first-order per-parameter adaptive optimizers in the Adam/Lion sense. Ano instead offers a simple, general optimizer that retains such efficiency while improving stability under noise and non-stationarity.

**Discussion.** Ano unifies sign-based and adaptive-moment ideas via a per-parameter direction–magnitude split (momentum for direction, raw gradients for scale), which we find particularly robust in high-variance regimes while remaining competitive on standard tasks.

## 3 ALGORITHM

The full Ano algorithm is summarized in Algorithm 1. Like Adam, it maintains first- and second-moment estimates $m_k, v_k$, but introduces two key innovations described below: one targeting the decoupling of update direction and magnitude, and the other improving variance adaptation under noisy gradients.

Ano algorithm is presented below:

---

**Algorithm 1:** Ano Algorithm

---

**Input:** Initial parameters $x_1 \in \mathbb{R}^d$, learning rate $\eta_k$, betas decay rates $\beta_1, \beta_2 \in [0, 1)$, $\epsilon > 0$, weight decay $\lambda$

Initialize $m_0 = 0$, $v_0 = 0$
**for** $k = 1$ **to** $K$ **do**
    Compute gradient $g_k = \nabla \ell(x_k)$
    $m_k = \beta_1 m_{k-1} + (1 - \beta_1) g_k$
    $v_k = \beta_2 v_{k-1} - (1 - \beta_2) \cdot \text{sign}(v_{k-1} - g_k^2) \cdot g_k^2$
    $\hat{v_k} = \frac{v_k}{1 - \beta 2^k}$
    $x_{k+1} = x_k - \frac{\eta_k}{\sqrt{\hat{v_k}} + \epsilon} \cdot |g_k| \cdot \text{sign}(m_k) - \eta_k \lambda x_k$

---

**Sign–Magnitude Decoupling.** We explicitly decouple the direction and magnitude of parameter updates to mitigate the conservative dynamics of Adam. In Adam, both signals are derived from the momentum term $m_k$, so when large noise spikes occur, their opposing effects can partially cancel out, reducing the effective momentum and thereby slowing down the updates. Ano keeps the direction $\text{sign}(m_k)$ for robustness to noise but replaces the momentum magnitude with the instantaneous gradient norm $|g_k|$ for a better scaling.

Concretely, recall that Adam updates parameters via

$$x_{k+1} = x_k - \frac{\eta_k}{\sqrt{v_k} + \epsilon} \cdot m_k = x_k - \frac{\eta_k}{\sqrt{\hat{v_k}} + \epsilon} \underbrace{|m_k|}_{\text{magnitude}} \cdot \underbrace{\text{sign}(m_k)}_{\text{direction}}.$$

Our optimizer **Ano** performs the same directional move but replaces the momentum magnitude with $|g_k|$:

$$x_{k+1} = x_k - \frac{\eta_k}{\sqrt{\hat{v_k}} + \epsilon} \underbrace{|g_k|}_{\text{magnitude}} \cdot \underbrace{\text{sign}(m_k)}_{\text{direction}}.$$

**Second-Moment Term.** Ano improves variance dynamics for stability and fast recovery in particular for non-stationary landscape. Adam's exponential moving average (Kingma & Ba, 2015) keeps noise spikes alive for many iterations, inflating the variance estimate and shrinking steps even after the signal improves. Yogi (Zaheer et al., 2018) addresses this with asymmetric updates for faster decay. We extend Yogi by introducing a decay factor that explicitly controls variance memory, maintaining the exponential structure while allowing smooth forgetting of outdated information. This mechanism naturally assigns greater weight to recent gradients, thereby enhancing adaptation in dynamic environments, which is essential in non-stationary environments. Formally,

$$v_k = \beta_2 v_{k-1} - (1 - \beta_2)\,\text{sign}(v_{k-1} - g_k^2)\,g_k^2,$$

turning the variance term into a memory-controlled statistic rather than a purely reactive estimate.

**Bias Correction and Weight Decay.** Since Ano relies solely on the momentum direction for updates, bias correction of its magnitude is unnecessary and omitted for simplicity (same as Lion) but keep it for the variance estimate. Weight decay follows AdamW (Loshchilov & Hutter, 2019) for decoupled regularization.

**Hyperparameters.** Like Adam, Ano maintains first and second moments estimators : $m_k$ and $v_k$, each regulated by decay rates $\beta_1$ and $\beta_2$, with $\beta_1 \in [0, 1)$ and $\beta_2 \in [\frac{1}{2}, 1)$. We set $\beta_1 = 0.92$ and $\beta_2 = 0.99$ for stable convergence. Additionally, a weight decay coefficient $\lambda \in [0, +\infty)$ is employed to mitigate overfitting.

## 4 EXTENSION

Inspired by our convergence analysis, we extend Ano to include a time-dependent momentum parameter $\beta_1$, resulting in a variant we denote **Anolog** (Ano with logarithmic scheduling). While Ano consistently yields the best raw performance, Anolog provides a practical advantage by removing the need to tune $\beta_1$. This reduction in hyperparameter sensitivity makes Anolog a competitive choice in scenarios with limited tuning budgets, despite its slightly lower peak performance.

We define $\beta_{1,k} = 1 - \frac{1}{\log(k+2)}$, motivated by both theoretical considerations and empirical evidence favoring slow, progressive adjustments to optimization hyperparameters. A gradually increasing $\beta_1$ enlarges the effective averaging window of the momentum, thereby reducing the impact of stochastic gradient noise as training proceeds. In contrast, more aggressive schedules (e.g., square-root) may render the momentum insufficiently responsive to recent gradient information, particularly in non-stationary settings where rapid adaptation is crucial. Section 7 provides empirical and ablation results comparing this logarithmic schedule against square-root ($\beta_{1,k} = 1 - \frac{1}{\sqrt{k}}$) and harmonic ($\beta_{1,k} = 1 - \frac{1}{k}$) schedules.

Full Anolog pseudo code can be found in Appendix A - Algorithm 2.

## 5 ANALYSIS

### 5.1 THEORETICAL ANALYSIS

We provide non-asymptotic convergence guarantees for **Ano** under standard assumptions commonly used in adaptive stochastic optimization (Kingma & Ba, 2015; Reddi et al., 2018). Consider the stochastic optimization problem: $\min_{x \in \mathbb{R}^d} f(x)$, where $f$ is differentiable, $L$-smooth, and bounded below. Let $g_{k,i}$ denote the $i$-th coordinate of the stochastic gradient at iteration $k$ and $\mathcal{F}_{k-1}$ the filtration k-1. We assume that the gradient is bounded, $|\nabla_i f(x_k)| \leq G, \forall x \in \mathbb{R}^d$, the stochastic gradient is unbiased ($\mathbb{E}[g_{k,i} \mid \mathcal{F}_{k-1}] = \nabla_i f(x_k)$), and the variance is bounded ($\mathbb{E}[(g_{k,i} - \nabla_i f(x_k))^2 \mid \mathcal{F}_{k-1}] \leq \sigma^2$).

**Main result.** Following recent convergence analyses of sign-based optimizers, especially Lion (Dong et al., 2024) and SignSGD (Sun et al., 2023), we assume a learning-rate schedule $\eta_k = \eta/k^{3/4}$ and $\beta_{1,k} = 1 - 1/\sqrt{k}$, the iterates generated by **Ano** satisfy:

$$\min_{0 \leq k < K} \mathbb{E}[\|\nabla f(x_k)\|^2] = \mathcal{O}(K^{-1/4} \log K) = \tilde{\mathcal{O}}(K^{-1/4}),$$

up to logarithmic factors, in the general non-convex stochastic setting.

**Proof sketch.** Using a sign-mismatch lemma (Lemma 2, Appendix D), we show that the probability of momentum–gradient disagreement decays as $\mathcal{O}(1/\sqrt{k})$. Then, using $L$-smoothness and the previous lemma, we establish the inequality:

$$\mathbb{E}[f(x_{k+1})] \leq \mathbb{E}[f(x_k)] - \frac{\eta_k}{\tilde{G} + \varepsilon} \mathbb{E}[\|\nabla f(x_k)\|^2] + \mathcal{O}\left(\frac{\eta_k}{k^{1/4}}\right) + \mathcal{O}(\eta_k^2),$$

where the last two terms represent, respectively, stochastic noise and the adaptivity of the step size.

**Discussion.** Our bound matches those recently established for sign-based optimizers such as *Lion*(Dong et al., 2024) and *Signum*(Bernstein et al., 2018), while relying on less restrictive assumptions (e.g., no requirement for growing batch sizes). Compared to adaptive schemes (SGD, Adam, Yogi) achieving $\mathcal{O}(K^{-1/2})$, our $\tilde{\mathcal{O}}(K^{-1/4})$ rate stems from a fundamental limitation of sign-based methods: ensuring stable updates requires decaying step sizes $\eta_k = \mathcal{O}(k^{-3/4})$ which, in turn, constrains the overall convergence rate. Full proofs are in Appendix D.

### 5.2 NOISE ROBUSTNESS ANALYSIS

We assess noise robustness by training a CIFAR-10 CNN adding Gaussian noise $g_k \leftarrow g_k + \mathcal{N}(0, \sigma^2)$ into every mini-batch gradient before the optimizer update (Alex, 2009). We vary only the noise level $\sigma$ over five values, keep each optimizer's default $\beta$ and recommended learning rate (Full hyperparameters tabs can be found in Appendix C) for a computer vision task and report mean test accuracy over 5 seeds.[1]

| Optimizer | $\sigma = 0$ | 0.01 | 0.05 | 0.10 | 0.20 |
|-----------|----------|------|------|------|------|
| Ano | 82.10 | 78.71 | 70.88 | 65.93 | 59.54 |
| Adam | 80.67(**-1.43**) | 75.97(**-2.74**) | 66.86(**-4.02**) | 60.83(**-5.10**) | 52.46(**-7.08**) |
| Lion | 81.04(**-1.05**) | 77.80(**-0.91**) | 69.62(**-1.26**) | 64.02(**-1.91**) | 56.82(**-2.72**) |
| Grams | 71.34(**-10.76**) | 77.90(**-0.81**) | 70.57(**-0.31**) | 65.47(**-0.46**) | 58.80(**-0.74**) |

Table 1: CIFAR-10 test accuracy (%). Numbers in parentheses indicate the gap (percentage points) relative to *Ano*.

The performance gap between Ano/Adam and Ano/Lion widens with noise magnitude, reaching a $-6.8$-point advantage at $\sigma = 0.20$ for Adam and a $-2.7$-point advantage for Lion (Table 1). Another

---

[1] 95%CI omitted here for readability. The full table with 95% CI is available in Appendix E - tab 9

noteworthy observation is that Grams improves with a small injected noise ($\sigma = 0.01$). We hypothesize that this injected perturbation amplifies short-term oscillations, enlarging its second-moment (variance) estimate and thereby shrinking the step size, allowing Grams to refine its iterates more cautiously in a noisy landscape. Overall, these results support our central claim that decoupling update direction from magnitude stabilizes learning under high variance, avoiding the over-smoothing of momentum-coupled schemes.

## 6 EXPERIMENTS

We evaluate **Ano** and **Anolog** across three domains: computer vision (CV), natural language processing (NLP), and deep reinforcement learning (DRL). Our experimental goal is deliberately asymmetric. Ano was designed for highly noisy and non-stationary regimes—most notably reinforcement learning—and this is where we expect its benefits to appear. In contrast, CV and NLP experiments serve as *diagnostic checks*: they assess whether Ano behaves sensibly in stable, low-noise supervised settings, without claiming superiority over the optimizers that dominate large-scale vision or language training.

Hyperparameters for all methods are selected through per-domain proxy searches (DRL, CV, NLP), each allocated a fixed 40 GPU-hour budget centered on literature defaults (Appendix C). Final results are averaged over 5 seeds for CV/NLP and 10 seeds for DRL. Full search spaces and selected configurations are provided in Appendices C–F.1.

### 6.1 COMPUTER VISION

Computer vision provides a well-established testbed for optimization, but modern CIFAR and ImageNet pipelines are highly stable and low-noise. As such, we use CIFAR-100 (Alex, 2009) with ResNet-34 (He et al., 2016) primarily to verify that Ano remains competitive and does not degrade in a regime *outside* its intended scope.

**CIFAR-100.** We use the standard CIFAR augmentation (random crop with 4-pixel padding + horizontal flip), following Zagoruyko & Komodakis (2016).

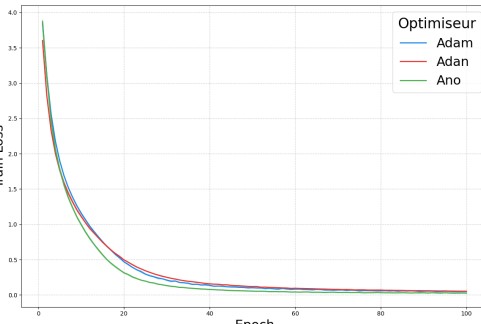

| Optimizers | Test Accuracy | Training Loss |
|---|---|---|
| *Default* | | |
| Adam | $69.57 \pm 0.22$ | 0.037 |
| Adan | $69.87 \pm 0.09$ | 0.049 |
| Lion | $66.25 \pm 0.61$ | 0.064 |
| Grams | $68.33 \pm 0.40$ | 0.045 |
| Ano | $\mathbf{70.31 \pm 0.50}$ | **0.015** |
| Anolog | $64.84 \pm 1.19$ | 0.019 |
| | | |
| *Tuned* | | |
| Adam | $69.61 \pm 0.23$ | 0.042 |
| Adan | $69.09 \pm 0.16$ | 0.049 |
| Lion | $68.77 \pm 0.26$ | 0.048 |
| Grams | $68.11 \pm 0.27$ | 0.048 |
| Ano | $\mathbf{69.89 \pm 0.42}$ | **0.022** |
| Anolog | $68.41 \pm 0.58$ | **0.032** |

Figure 1: Training loss on CIFAR-100. Ano reduces loss faster and more stably than Adam.

Table 2: Test accuracy and training loss on CIFAR-100.

Table 2 reports accuracy and training loss under both default and tuned settings. Ano performs slightly better than Adam and Adan, but—as expected for a saturated, low-noise benchmark—the margins remain modest and largely within the range observed for modern optimizers. The qualitative takeaway is that Ano behaves as a stable supervised optimizer, without harming convergence or accuracy.

### 6.2 NATURAL LANGUAGE PROCESSING

As a cornerstone of modern artificial intelligence, natural language processing (NLP) warrants careful evaluation of optimization algorithms. As with Computer Vision, the purpose of these experiments is not to position Ano as a competitive alternative in large-scale pretraining, but to verify that

its behavior remains consistent across supervised domains and to highlight small-scale noisy tasks where gradient variance plays a significant role.

**GLUE** We evaluate Ano on the GLUE benchmark (Wang et al., 2019), covering eight classification tasks (excluding WNLI). All experiments finetune the public `bert-base-uncased` model (Devlin et al., 2019) with a sequence length of 128, batch size 32, weight decay 0.01, a linear schedule with 10% warmup, mixed precision, and 3 epochs (5 for smaller tasks: CoLA, MRPC, RTE, STS-B), using 5 seeds.

| Optimizer | CoLA | MNLI | MRPC | QNLI | QQP | RTE | SST-2 | STS-B | Average |
|---|---|---|---|---|---|---|---|---|---|
| *Default* | | | | | | | | | |
| Adam | **59.40 ± 1.67** | **84.62 ± 0.10** | 88.06 ± 0.82 | **91.60 ± 0.15** | 89.64 ± 0.10 | 66.67 ± 1.59 | 92.73 ± 0.46 | 88.44 ± 0.27 | 82.64 |
| Adan | 55.65 ± 0.53 | 84.17 ± 0.07 | 84.40 ± 0.83 | 91.10 ± 0.14 | 88.85 ± 0.04 | 61.49 ± 1.30 | 92.02 ± 0.20 | 87.26 ± 0.63 | 80.62 |
| Lion | 57.76 ± 1.76 | 83.76 ± 0.23 | 87.13 ± 0.81 | 90.63 ± 0.68 | 89.46 ± 0.05 | 62.89 ± 1.17 | 91.67 ± 0.52 | 88.00 ± 0.25 | 81.41 |
| Grams | 56.15 ± 0.92 | 83.89 ± 0.11 | 84.92 ± 0.60 | 91.10 ± 0.04 | 88.48 ± 0.08 | 63.36 ± 1.60 | 92.34 ± 0.19 | 87.57 ± 0.32 | 80.98 |
| **Ano (Ours)** | 58.36 ± 1.15 | 84.33 ± 0.17 | **88.96 ± 0.50** | 91.25 ± 0.46 | **89.71 ± 0.11** | **69.25 ± 2.94** | **92.80 ± 0.41** | 88.70 ± 0.12 | **82.92** |
| **Anolog (Ours)** | 57.07 ± 2.41 | 84.55 ± 0.09 | 88.26 ± 0.76 | 91.51 ± 0.10 | **89.71 ± 0.12** | 67.87 ± 1.94 | 92.75 ± 0.15 | **88.95 ± 0.32** | 82.58 |
| *Tuned* | | | | | | | | | |
| Adam | 57.66 ± 2.39 | 84.18 ± 0.16 | 88.09 ± 0.79 | 91.12 ± 0.17 | 89.55 ± 0.08 | 68.47 ± 2.91 | 92.18 ± 0.08 | 88.76 ± 0.36 | 82.50 |
| Adan | 57.71 ± 0.92 | **84.84 ± 0.10** | 88.14 ± 0.40 | 91.71 ± 0.21 | **89.78 ± 0.07** | 65.40 ± 1.66 | 92.68 ± 0.26 | 88.57 ± 0.44 | 82.35 |
| Lion | 56.30 ± 0.55 | 82.38 ± 0.06 | 86.83 ± 2.91 | 90.36 ± 0.42 | 88.60 ± 0.13 | 63.75 ± 5.50 | 91.47 ± 0.24 | 88.58 ± 0.40 | 81.03 |
| Grams | 58.18 ± 1.12 | 84.64 ± 0.11 | **89.05 ± 0.36** | **91.79 ± 0.17** | 89.66 ± 0.06 | 67.22 ± 2.55 | **92.98 ± 0.31** | 88.53 ± 0.26 | 82.76 |
| **Ano (Ours)** | **58.51 ± 0.75** | 84.39 ± 0.12 | 88.53 ± 1.14 | 91.30 ± 0.48 | 89.73 ± 0.07 | **69.25 ± 3.01** | 92.66 ± 0.14 | 88.74 ± 0.11 | **82.89** |
| **Anolog (Ours)** | 57.07 ± 2.41 | 84.55 ± 0.09 | 88.26 ± 0.76 | 91.51 ± 0.10 | 89.71 ± 0.12 | 67.87 ± 1.94 | 92.75 ± 0.15 | **88.95 ± 0.32** | 82.58 |

Table 3: Average performance (mean ± CI95%) of different optimizers on GLUE benchmark tasks.

As shown in Table 3, Ano and Anolog match or slightly surpass strong baselines on average. The improvements are concentrated on small, noisy tasks such as MRPC, CoLA, and RTE, where stochasticity in gradient signals is highest (see Appendix B.3). On large, stable tasks (MNLI, QQP, QNLI), all optimizers achieve similar performance, which aligns with the expectation that Ano offers limited benefits in stationary low-noise regimes.

## 6.3 DEEP REINFORCEMENT LEARNING

Reinforcement learning (RL) is characterized by high gradient variance and strong non-stationarity, both of which heavily influence optimizer behavior (Henderson et al., 2018; François-Lavet et al., 2018). Since these conditions align with the regime Ano is designed for, RL is where we expect the largest gains. For computational efficiency, hyperparameters are tuned on *HalfCheetah* using 100k-step runs. We acknowledge that this shorter horizon may favor slightly larger learning rates and thus may not perfectly reflect long-horizon behavior (e.g., at 1M steps). To reduce this bias, each baseline reports the better of its default or tuned configuration, ensuring no method is penalized by the tuning protocol. Tables explicitly indicate which setting was selected, and full results are provided in E. Following best practices (Henderson et al., 2018; Agarwal et al., 2021), we report IQM and 95

**Soft-Actor Critic.** In this section, we employ the Soft Actor-Critic (SAC) algorithm (Haarnoja et al., 2018) in the MuJoCo suite from the Gymnasium framework (Todorov et al., 2012; Towers et al., 2024). We reuse the standard SAC hyperparameter (full list in Appendix F.1 - Tab 13), as reported in the original work and subsequent studies and only vary the optimizer for actor, critics, and temperature. For each optimizer, we run 10 seeds, with 1M steps. We report below the average mean score on a 50-episodes test evaluation.

As summarized in Table 4, Ano performs favorably compared to Adam and other baselines across the MuJoCo tasks. On average, it achieves a **+10%** improvement in normalized score[2], both under default and tuned hyperparameters. Without tuning, Ano ranks first in 4 out of 5 tasks; with best version, it remains the top optimizer in 3 out of 5 tasks. Although not always the best performer, Ano consistently ranks among the strongest optimizers, with its scores typically within or close to the 95% confidence intervals of the best baselines. Figure 2 shows that Ano reaches the final performance of Adam using approximately 50–70% fewer training steps, except for *Humanoid*. To address potential concerns about hyperparameter tuning, we evaluated the sensitivity to learning rate and momentum coefficients on a 100k-step *HalfCheetah* proxy (Figure 3; see C for full details). Ano

---

[2]The normalized average is obtained by linearly rescaling each score between the minimum and maximum values observed across optimizers, followed by averaging. Complete normalized results are reported in Table 10.

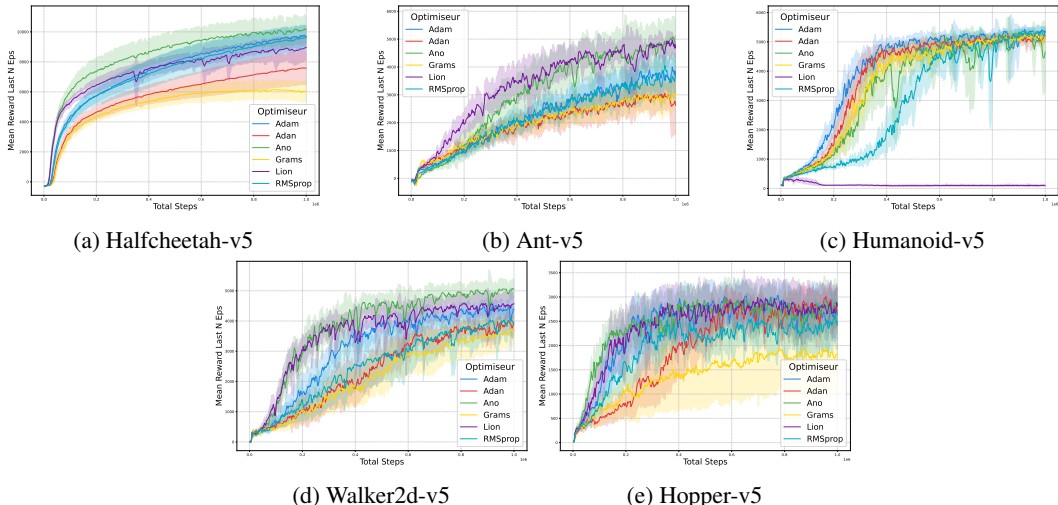

(a) Halfcheetah-v5      (b) Ant-v5      (c) Humanoid-v5

(d) Walker2d-v5      (e) Hopper-v5

Figure 2: Rewards over time for several MuJoCo environments, with baselines and 95% confidence intervals. The green curve corresponds to **Ano (ours)**.

| Optimizers | HalfCheetah | Ant | Humanoid | Walker2d | Hopper | Mean Rank | Norm. Avg |
|---|---|---|---|---|---|---|---|
| *Default* | | | | | | | |
| Adam | 10549.48 ± 721.55 | 4336.64 ± 698.72 | **5357.14 ± 211.97** | 4462.51 ± 588.77 | 3164.71 ± 600.48 | 3.4 | 90.66 |
| RMSprop | 10506.23 ± 852.19 | 4234.37 ± 763.65 | 5395.51 ± 126.80 | 4160.06 ± 480.62 | 2973.86 ± 571.05 | 5.6 | 87.83 |
| Adan | 7805.20 ± 1154.02 | 2985.19 ± 1018.79 | 5080.74 ± 305.26 | 4092.13 ± 379.92 | 3222.62 ± 235.25 | 5.6 | 78.38 |
| Lion | 9527.96 ± 805.42 | 4948.26 ± 243.05 | 98.22 ± 32.33 | 4612.63 ± 367.77 | 3087.27 ± 628.06 | 4.4 | 71.74 |
| Grams | 6782.60 ± 715.12 | 3207.30 ± 531.06 | 5104.10 ± 692.14 | 3656.66 ± 658.82 | 1475.34 ± 927.22 | 6.4 | 65.88 |
| **Ano (Ours)** | **10864.09 ± 1052.24** | **5285.44 ± 729.86** | 5255.62 ± 815.92 | **5227.86 ± 436.49** | **3535.32 ± 780.96** | **1.4** | **99.48** |
| **Anolog (Ours)** | 10557.05 ± 560.70 | 5089.12 ± 522.94 | 5242.78 ± 173.98 | 4606.02 ± 478.36 | 3314.12 ± 539.95 | 2.6 | 94.50 |
| *Best Version* | | | | | | | |
| Adam [Default] | 10549.48 ± 721.55 | 4336.64 ± 698.72 | **5357.14 ± 211.97** | 4462.51 ± 588.77 | 3164.71 ± 600.48 | 4.6 | 90.38 |
| RMSprop [Default] | 10506.23 ± 852.19 | 4234.37 ± 763.65 | 5395.51 ± 126.80 | 4160.06 ± 480.62 | 2973.86 ± 571.05 | 5.6 | 87.83 |
| Adan [Tuned] | 10822.40 ± 475.75 | 5239.69 ± 270.96 | 4792.62 ± 904.44 | 4686.83 ± 502.28 | 3514.42 ± 143.57 | 3.2 | 95.01 |
| Lion [Tuned] | 10482.06 ± 1018.86 | 4848.41 ± 821.79 | 1349.15 ± 1322.56 | 4876.76 ± 253.22 | **3592.87 ± 70.26** | 4.2 | 81.30 |
| Grams [Tuned] | 10533.70 ± 866.69 | 4607.59 ± 505.08 | 5147.04 ± 487.55 | 4644.45 ± 498.08 | 3147.82 ± 605.03 | 5.0 | 91.20 |
| **Ano (Ours) [Default]** | **10864.09 ± 1052.24** | **5285.44 ± 729.86** | 5255.62 ± 815.92 | **5227.86 ± 436.49** | 3535.32 ± 780.96 | **1.6** | **99.16** |
| **Anolog (Ours) [Default]** | 10557.05 ± 560.70 | 5089.12 ± 522.94 | 5242.78 ± 173.98 | 4606.02 ± 478.36 | 3314.12 ± 539.95 | 2.6 | 94.20 |

Table 4: Comparison of the IQM ± CI95% of different optimizers across environments.

shows lower sensitivity than Adam to both learning rate and betas, suggesting that its performance gains are not solely due to more favorable hyperparameter choices.

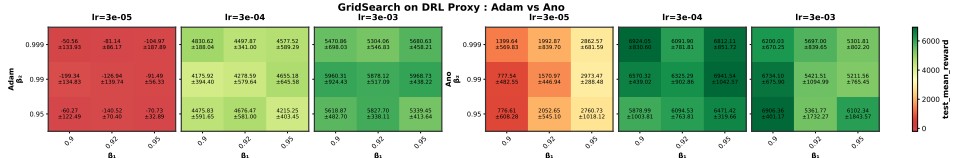

Figure 3: Hyperparameter robustness on a MuJoCo proxy (HalfCheetah with SAC). **Adam** on the left, **Ano(ours)** on the right.

**Proximal Policy Optimization.** To assess the generality of our results, we also evaluate a discrete-action PPO agent (Schulman et al., 2017) on the Atari Learning Environment (ALE) (Bellemare et al., 2013). For efficiency, we use the *Atari-5* subset from Aitchison et al. (Aitchison et al., 2023), which explains 98.4% of the full-suite variance. We rely on the CleanRL PPO implementation (Huang et al., 2022) with default model and optimization settings (full details in Appendix F.1, Tab. 14). Environments run via EnvPool (Weng et al., 2022). Observations are resized to 84×84, grayscaled, and stacked over 4 frames; we use action repeat 4, up to 30 no-ops on reset, sticky actions with probability 0.25 (Machado et al., 2018), FireReset when required, and the full action set. Training uses clipped rewards in $[-1, 1]$; evaluation uses unclipped returns. Agents train for 10M steps ($\approx$ 40M frames) and are evaluated every 200k steps over 50 episodes using the same wrappers

as training (minus reward clipping). We report the mean final score, normalized per environment using theoretical minima (-18 for DoubleDunk, 0 otherwise) and the maximum baseline score[3].

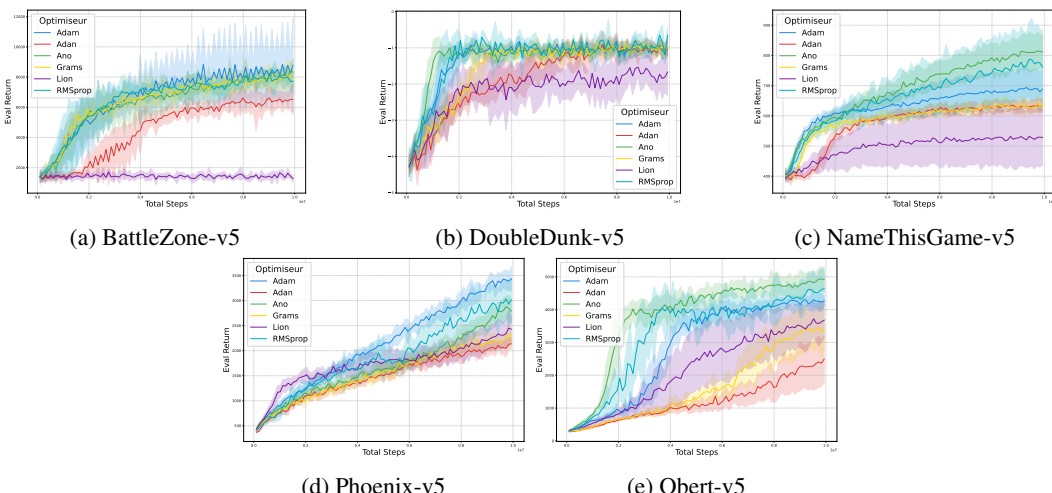

(a) BattleZone-v5      (b) DoubleDunk-v5      (c) NameThisGame-v5

(d) Phoenix-v5      (e) Qbert-v5

Figure 4: Rewards over time for Atari5 Benchmark, with baselines and 95% confidence intervals. The green curve corresponds to **Ano (ours)**.

| Optimizers | BattleZone-v5 | DoubleDunk-v5 | NameThisGame-v5 | Phoenix-v5 | Qbert-v5 | Mean Rank | Norm. Avg |
|---|---|---|---|---|---|---|---|
| *Default* | | | | | | | |
| Adam | $7615.00 \pm 1299.89$ | $-1.08 \pm 0.21$ | $665.35 \pm 64.89$ | $\mathbf{3443.30 \pm 240.68}$ | $4257.80 \pm 135.90$ | 4.4 | 87.54 |
| RMSprop | $7686.67 \pm 859.43$ | $\mathbf{-0.67 \pm 0.22}$ | $798.00 \pm 118.66$ | $3031.13 \pm 410.22$ | $4585.67 \pm 538.44$ | 2.4 | 90.09 |
| Adan | $6480.00 \pm 702.14$ | $-0.91 \pm 0.23$ | $638.35 \pm 18.08$ | $2106.90 \pm 110.74$ | $2665.00 \pm 708.69$ | 5.4 | 74.11 |
| Lion | $1392.00 \pm 139.66$ | $-1.46 \pm 0.65$ | $508.15 \pm 91.28$ | $2432.35 \pm 234.61$ | $3768.00 \pm 520.95$ | 6.8 | 61.36 |
| Grams | $7908.00 \pm 687.61$ | $-0.93 \pm 0.24$ | $633.80 \pm 23.52$ | $2234.40 \pm 130.73$ | $3670.12 \pm 634.75$ | 4.6 | 82.41 |
| **Ano (Ours)** | $\mathbf{8095.00 \pm 494.70}$ | $-0.97 \pm 0.14$ | $\mathbf{845.35 \pm 56.39}$ | $2813.00 \pm 243.04$ | $\mathbf{4828.80 \pm 386.73}$ | **2.2** | **95.99** |
| **Anolog (Ours)** | $7485.00 \pm 1010.66$ | $-0.98 \pm 0.14$ | $751.05 \pm 74.60$ | $2983.00 \pm 236.42$ | $4773.75 \pm 602.39$ | 3.6 | 93.00 |
| *Best Version* | | | | | | | |
| Adam [Baseline] | $7615.00 \pm 1299.89$ | $-1.08 \pm 0.21$ | $665.35 \pm 64.89$ | $\mathbf{3443.30 \pm 240.68}$ | $4257.80 \pm 135.90$ | 4.4 | 87.54 |
| RMSprop [Baseline] | $7686.67 \pm 859.43$ | $\mathbf{-0.67 \pm 0.22}$ | $798.00 \pm 118.66$ | $3031.13 \pm 410.22$ | $4585.67 \pm 538.44$ | 2.4 | 90.09 |
| Adan [Tuned] | $4840.00 \pm 2601.35$ | $-0.95 \pm 0.23$ | $754.20 \pm 51.44$ | $2647.20 \pm 534.22$ | $4524.75 \pm 448.74$ | 4.4 | 79.67 |
| Lion [Baseline] | $1392.00 \pm 139.66$ | $-1.46 \pm 0.65$ | $508.15 \pm 91.28$ | $2432.35 \pm 234.61$ | $3768.00 \pm 520.95$ | 6.8 | 61.36 |
| Grams [Tuned] | $7715.00 \pm 627.92$ | $-1.35 \pm 0.75$ | $690.40 \pm 76.27$ | $1989.15 \pm 201.64$ | $5049.25 \pm 624.43$ | 4.4 | 82.26 |
| **Ano (Ours) [Tuned]** | $\mathbf{8625.00 \pm 1870.44}$ | $-0.91 \pm 0.21$ | $\mathbf{828.10 \pm 67.66}$ | $2824.85 \pm 226.30$ | $\mathbf{5960.88 \pm 912.36}$ | **1.8** | **96.13** |
| **Anolog (Ours) [Baseline]** | $7485.00 \pm 1010.66$ | $-0.98 \pm 0.14$ | $751.05 \pm 74.60$ | $2983.00 \pm 236.42$ | $4773.75 \pm 602.39$ | 3.8 | 88.48 |

Table 5: Comparison of the IQM $\pm$ CI95% of different optimizers across Atari environments.

As shown in Table 5, Ano and RMSprop perform strongest overall among the baselines. In the default setting, their mean ranks are 2.2 and 2.4; with the best version, 1.8 and 2.4, respectively. Ano achieves the highest average normalized score and mean rank in both regimes, with approximately 6–7% higher normalized average than RMSprop and 10% higher than Adam. Notably, Ano outperforms Adam on *BattleZone*, *Name This Game*, and *Q*bert*, whereas Adam and RMSprop perform best on *Phoenix*. For *DoubleDunk*, all optimizers (except Lion) plateau at similar levels (Fig. 4b), so no clear winner emerges.

## 7 ABLATION STUDY

We conduct ablation studies on Ano and its variant Anolog to quantify the contribution of each design component and to justify using a logarithmic momentum schedule rather than the theoretically motivated square-root schedule. Table 6 summarizes all ablated variants. To provide a comprehensive evaluation, we compare performance on four benchmarks: *HalfCheetah* from MuJoCo (Todorov et al., 2012), CIFAR100 (Alex, 2009), and two tasks from the GLUE benchmark: the small and noisy MRPC task, and the larger, more stable SST2 task. We follow the same experimental protocols as in Section 6, except that for *HalfCheetah* we train for 500k steps.

---

[3]We normalize each environment score using its theoretical minimum and the maximum baseline score, then report the mean normalized score.

| Optimizer | Second Mom. Rule | Grad. Norm | Mom. Norm | Mom. Dir. | Decoup. WD | $\beta_{1,k}$ | Score DRL | Acc. (%) CIFAR-100 | Acc. (%) MRPC | Acc. (%) SST-2 |
|---|---|---|---|---|---|---|---|---|---|---|
| *Ano ablation* | | | | | | | | | | |
| Adam | Adam | ✗ | ✓ | ✓ | ✓ | $\beta_1$ | $7480.55 \pm 1323.36$ | $69.84 \pm 0.22$ | $85.93 \pm 0.92$ | $\mathbf{93.03 \pm 0.30}$ |
| YogiTweaked | Yogi+$\beta_2$-decay | ✗ | ✓ | ✓ | ✓ | $\beta_1$ | $8540.52 \pm 671.22$ | $68.62 \pm 2.36$ | $85.25 \pm 1.22$ | $92.75 \pm 0.32$ |
| Grams | Adam | ✗ | ✓ | ✗ | ✓ | $\beta_1$ | $5567.12 \pm 782.37$ | $70.20 \pm 0.17$ | $82.25 \pm 0.74$ | $92.29 \pm 0.23$ |
| YogiSignum | Yogi+$\beta_2$-decay | ✗ | ✗ | ✓ | ✓ | $\beta_1$ | $-285.58 \pm 41.11$ | $3.99 \pm 2.01$ | $68.38 \pm 0.00$ | $50.92 \pm 0.00$ |
| Signum | ✗ | ✗ | ✗ | ✓ | ✓ | $\beta_1$ | $9393.64 \pm 1399.78$ | $65.11 \pm 0.90$ | $86.42 \pm 0.72$ | $90.41 \pm 0.30$ |
| SignumGrad | ✗ | ✓ | ✗ | ✓ | ✓ | $\beta_1$ | $-$ | $53.93 \pm 0.68$ | $68.38 \pm 0.00$ | $53.33 \pm 2.62$ |
| AdamGrad | Adam | ✓ | ✗ | ✓ | ✓ | $\beta_1$ | $9855.19 \pm 1173.19$ | $70.30 \pm 0.38$ | $86.96 \pm 0.85$ | $92.71 \pm 0.45$ |
| AnoWoTweak | Yogi | ✓ | ✗ | ✓ | ✓ | $\beta_1$ | $9053.10 \pm 792.13$ | $\mathbf{70.32 \pm 1.20}$ | $\mathbf{87.06 \pm 0.69}$ | $92.80 \pm 0.45$ |
| **Ano** | Yogi+$\beta_2$-decay | ✓ | ✗ | ✓ | ✓ | $\beta_1$ | $\mathbf{10520.00 \pm 416.07}$ | $69.74 \pm 0.45$ | $86.76 \pm 0.63$ | $92.52 \pm 0.31$ |
| *Anolog ablation* | | | | | | | | | | |
| Anoall | Yogi+$\beta_2$-decay | ✓ | ✗ | ✓ | ✓ | $1 - 1/k$ | $-221.45 \pm 22.25$ | $29.48 \pm 2.40$ | $68.38 \pm 0.00$ | $52.22 \pm 1.88$ |
| Anosqrt | Yogi+$\beta_2$-decay | ✓ | ✗ | ✓ | ✓ | $1 - 1/\sqrt{k}$ | $8750 \pm 860.50$ | $\mathbf{67.26 \pm 0.41}$ | $\mathbf{86.18 \pm 1.08}$ | $91.74 \pm 0.53$ |
| **Anolog** | Yogi+$\beta_2$-decay | ✓ | ✗ | ✓ | ✓ | $1 - 1/\log k$ | $\mathbf{9472.73 \pm 968.26}$ | $67.00 \pm 0.80$ | $85.25 \pm 1.79$ | $\mathbf{92.78 \pm 0.16}$ |

Table 6: Ablation of our proposed optimizer(**Ano**) and its extension (**Anolog**). Columns on the left indicate which components are active; columns on the right report mean performance $\pm$ 95% CI

As shown in Table 6, **Ano** achieves the highest mean return in deep reinforcement learning, improving by roughly 7% over the same algorithm with Adam-style second moments and about 15% over Ano with Yogi-style second moments, while staying within 1% of the best accuracy on all supervised learning tasks. Using only the sign of the momentum (e.g., Signum, AdamGrad, AnoWoDecay) also improves DRL performance, supporting our design choice to decouple sign and magnitude: this enables larger update steps, which are particularly beneficial in noisy or non-stationary environments. Performance drops when either gradient normalization (SignumGrad) or gradient magnitude (YogiSignum) is removed, underscoring their complementary roles. For momentum schedules, the logarithmic schedule improves DRL return over the $\sqrt{k}$ schedule while staying within the 95% confidence interval on other tasks, motivating its inclusion in the final design.

## 8 LIMITATIONS AND DISCUSSION

Through our design and empirical analysis of Ano, we identified three main limitations: First, the choice of $\beta_2$-decay appears particularly beneficial in reinforcement learning or highly non-stationary loss landscapes, where older gradients can be misleading and a rapid adaptation is crucial. However, in more stationary settings, such as classical supervised learning in CV and NLP, we observed that the variance estimate in vanilla Yogi often leads to more stable and effective training. Our focus on noisy, non-stationary environments motivates this design choice, though we acknowledge that its relevance to more conventional settings remains an open question. Second, by construction, Ano favors larger step sizes to improve reactivity. While this design is advantageous in non-stationary contexts, it can also introduce instability. For example, our experiments with Nesterov-style acceleration, inspired by Adan, amplified rather than mitigated this issue. Third, our experiments on classical CV and NLP tasks remain limited in scale, as Ano was primarily designed for highly non-stationary and noisy environments. In more stationary settings with longer training horizons, we observed that Adam can sometimes achieve better stability due to its smaller update steps. While these results suggest that Ano's benefits are not restricted to DRL, assessing its relevance to large-scale CV or NLP tasks lies beyond the current scope and is left for future work.

## 9 CONCLUSION

We introduced **Ano**, an alternative to momentum-based adaptive optimizers that decouples direction and magnitude to improve robustness in noisy and non-stationary settings. Under standard smoothness and bounded-noise assumptions, we derive non-asymptotic guarantees comparable to existing analyses of sign-based methods (e.g., Signum, Lion) under similar decay schedules. Empirically, Ano achieves notable improvements in reinforcement learning and noisy NLP tasks while remaining competitive on low-noise benchmarks. Future work will focus on developing variance estimators tailored to supervised learning, integrating Nesterov-style look-ahead, benchmarking on MARL benchmark, and enhancing stability in long, stationary training regimes.[4]

---

[4]LLMs (GPT, Gemini) were used for minor text editing, LaTeX formatting, informal feedback on early drafts, and retrieval of related work and references; all research ideas, analyses, and conclusions remain solely those of the authors.

## 10 REPRODUCIBILITY STATEMENT

All datasets used in this work are publicly available. The full source code, including training, pre-processing, and result visualization scripts, as well as all experiment logs, is released in an anonymous repository[5]. The optimizer is also available as a pip package (PyTorch, TensorFlow, JAX) to facilitate implementation, but it isn't include in the source code for double bind review. Data preprocessing details, hyperparameter grids, and training protocols are described in Section 6 and provided in the source code. All experiments were run with fixed random seeds on a workstation with an RTX 5090 GPU and an Intel Core Ultra 9 CPU using CUDA 12.9 and PyTorch 2.9.0.

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

## A  ANOLOG PSEUDO CODE

---

**Algorithm 2:** Anolog

---

**Input:** Initial parameters $x_1 \in \mathbb{R}^d$, learning rate $\eta_k$, betas decay rate $\beta_2 \in [0, 1)$, weight decay $\lambda$, $\epsilon > 0$

Initialize $m_0 = 0$, $v_0 = 0$
**for** $k = 1$ **to** $K$ **do**

  Compute gradient $g_k = \nabla \ell(x_k)$
  $\beta_1 = 1 - \frac{1}{\log(k+2)}$
  $m_k = \beta_1 m_{k-1} + (1 - \beta_1) g_k$
  $v_k = \beta_2 v_{k-1} - (1 - \beta_2) \cdot \text{sign}(v_{k-1} - g_k^2) \cdot g_k^2$
  $\hat{v_k} = \frac{v_k}{1 - \beta_2^k}$
  $x_{k+1} = x_k - \frac{\eta_k}{\sqrt{\hat{v_k}} + \epsilon} \cdot |g_k| \cdot \text{sign}(m_k) - \eta_k \lambda x_k$

---

## B  ADDITIONNAL EXPERIMENTS

### B.1  NON-STATIONARY ANALYSIS

We assess the behavior of the four optimizers (Lion, Grams, Adam, and Ano) in a controlled non-stationary synthetic setting, we evaluated tracking performance under drifting objectives. We consider a switching quadratic sequence $f_t(x) = \frac{1}{2}\|x - \mu_t\|^2$, where the target $\mu_t$ is abruptly reassigned every 1000 iterations. This generates a piecewise-stationary landscape that isolates the effect of non-stationarity from model- or data-dependent factors. All optimizers are run with identical hyper-parameters (learning rate, momentum coefficients, etc.), and results are averaged over three random seeds. We report the resulting tracking error $\|x_t - \mu_t\|$ across time.

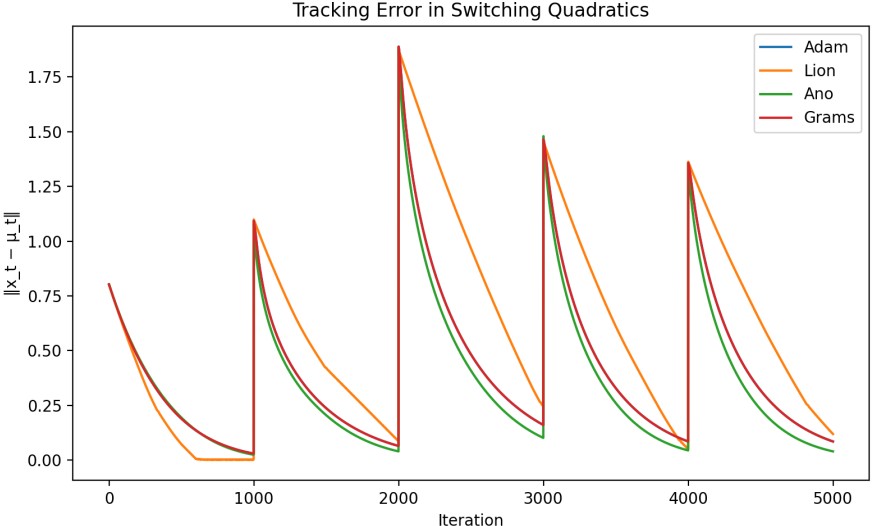

Figure 5: Tracking error $\|x_t - \mu_t\|$ in the switching quadratic setting (mean over 5 seeds). Vertical jumps correspond to resets of the target $\mu_t$.

Figure 5 shows that all methods exhibit the characteristic sawtooth pattern induced by the periodic switches. Grams and Adam produce nearly identical trajectories, reflecting similar sensitivity to abrupt shifts despite their different update formulations. Lion consistently recovers more slowly after each jump, leading to higher transient and steady-state errors. Ano adapts the fastest overall, rapidly reducing the post-switch deviation and achieving the lowest tracking error throughout the experiment.

### B.2 SENSITIVITY TO BUFFER SIZE AND BATCH SIZE

We additionally study the sensitivity of the underlying RL algorithm, and in particular SAC, to two key replay parameters: buffer size and batch size. Our objective is to assess whether the performance gains observed with our optimizer persist under different replay configurations. To this end, we perform a controlled ablation on HalfCheetah over 500k environment steps, using five random seeds and identical SAC hyperparameters across all runs. We report the InterQuartile Mean (IQM) along with the 95% confidence interval.

Table 7: Sensitivity study on buffer size and batch size. Metric: IQM $\pm$ CI95. $\Delta$ columns report Ano – Adam (absolute and relative).

| (A) Varying Buffer Size (Batch Size = 256) | | | | |
|---|---|---|---|---|
| Buffer Size | Adam | Ano | $\Delta$ (abs) | $\Delta$ (%) |
| $1 \times 10^5$ | $10083.7 \pm 1810.3$ | $11756.0 \pm 2383.6$ | +1672.3 | +16.6% |
| $2.5 \times 10^5$ | $8943.9 \pm 2024.5$ | $11174.1 \pm 2234.6$ | +2230.2 | +24.9% |
| $1 \times 10^6$ | $8265.4 \pm 2382.7$ | $10789.3 \pm 2090.8$ | +2523.9 | +30.5% |
| (B) Varying Batch Size (Buffer Size = $10^6$) | | | | |
| Batch Size | Adam | Ano | $\Delta$ (abs) | $\Delta$ (%) |
| 64 | $8107.2 \pm 1469.2$ | $9216.3 \pm 1221.8$ | +1109.2 | +13.7% |
| 256 | $8265.4 \pm 2382.7$ | $10789.3 \pm 2090.8$ | +2523.9 | +30.5% |
| 512 | $8477.2 \pm 1015.0$ | $10876.0 \pm 1066.6$ | +2398.8 | +28.3% |

The effect of batch size is relatively consistent across optimizers: both Adam and Ano show similar qualitative trends as the batch size increases, with Ano maintaining a stable advantage but no major change in relative sensitivity. In contrast, the buffer-size ablation reveals a clear difference in robustness. Adam's performance drops substantially as the replay buffer grows, indicating a strong dependence on the recency and distribution of stored transitions. Ano, however, remains considerably more stable across all buffer sizes, showing only mild degradation and preserving a sizeable performance margin. This suggests that Ano is markedly less sensitive to replay-buffer configuration, maintaining reliable learning even when long-range replay dynamics introduce distributional drift.

### B.3 GNS ACROSS DIFFERENT LANDSCAPES

To better understand when Ano provides the greatest benefit, we examine the mean Gradient Noise Scale (GNS), following the definition of McCandlish et al. (2018). The GNS provides a coarse estimate of the stochasticity of the optimization landscape: higher values indicate noisier gradients and larger variance across updates. As a practical guideline, we recommend considering Ano when the GNS exceeds roughly $10^3$; the higher the GNS, the more advantageous Ano tends to become in practice.

| Atari | Mean GNS |
|---|---|
| BattleZone | 7697 |
| DoubleDunk | 13971 |
| NameThisGame | 12084 |
| Qbert | 14151 |
| Phoenix | 7638 |
| **Mujoco** | |
| Humanoid-QOpt | 133322 |
| Humanoid-ActorOpt | 3956 |
| HalfCheetah-QOpt | 25530 |
| HalfCheetah-ActorOpt | 3331 |
| Ant-QOpt | 11366 |
| Ant-ActorOpt | 2103 |
| Hopper-QOpt | 15689 |
| Hopper-ActorOpt | 2395 |
| Walker2d-QOpt | 12502 |
| Walker2d-ActorOpt | 2517 |
| **GLUE** | |
| QNLI | 535 |
| STSB | 329 |
| QQP | 520 |
| MRPC | 1382 |
| SST2 | 744 |
| MNLI | 462 |
| RTE | 988 |
| COLA | 735 |

Table 8: Mean GNS values grouped by domain: Atari, Mujoco, and GLUE.

The GNS values (Table 8) reveal a clear separation across domains. Atari and the critic components of Mujoco tasks exhibit extremely high noise levels, often above $10^4$, placing them well within the regime where Ano provides substantial gains. Actor updates in Mujoco remain noisier than supervised NLP tasks but are generally closer to the $10^3$ threshold, suggesting more moderate benefits. In contrast, GLUE tasks show GNS values below $10^3$ for most datasets, indicating substantially lower stochasticity and a landscape in which Ano is less critical.

## C  Hyperparameter Tuning Protocol

To ensure a fair comparison, we conducted exhaustive grid searches over learning rates and momentum parameters on lightweight proxy tasks representative of each domain, using 5 independent seeds for each hyperparameter combination. In total, the campaign involved **2115** independent training runs: roughly 35 hours per optimizer when searching across three hyperparameters, and about 12 hours for optimizers with only two (e.g., Anolog, RMSprop). For Adan, which introduces a third momentum coefficient, we maintained a uniform computational budget by varying only $(\beta_1, \beta_3)$, that controlling the first and second moment estimates, while fixing the Nesterov term at its default value, $\beta_2 = 0.92$. For Anolog and RMSprop, we tuned only the learning rate and the variance term decay parameter ($\beta_2$ for Anolog and $\alpha$ for RMSprop). We then selected the configuration achieving the highest validation accuracy per seed, with final hyperparameters for all optimizers summarized in Table 12.

**Computer-vision proxy.**  From CIFAR-10 (Alex, 2009), we drew a balanced subset of 10,000 training images and 2,000 test images. We applied the identical augmentation pipeline used in Section 6 (random cropping, horizontal flips, and Cutout). Each hyperparameter configuration was trained for 20 epochs under five independent seeds with a ResNet-18 backbone(He et al., 2016).

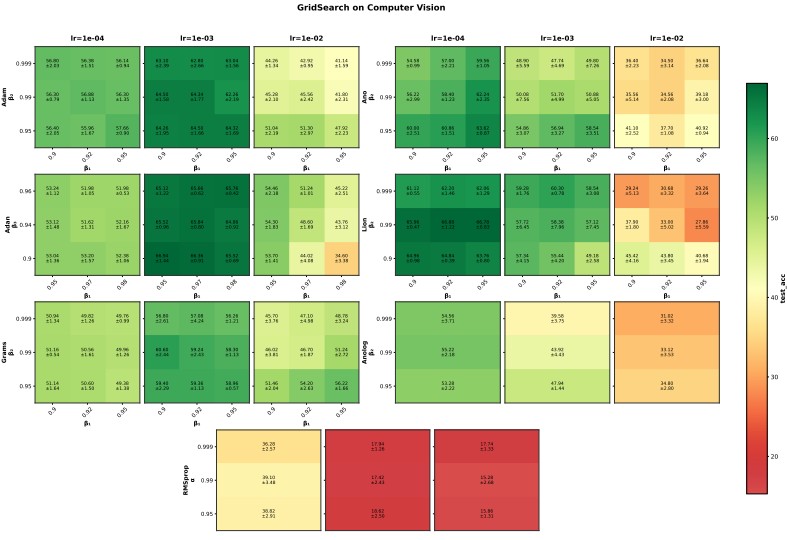

Figure 6: Grid search on the CIFAR-10 proxy (ResNet-18) for the optimizers.

**NLP proxy.**  We fine-tune Bert (Devlin et al., 2019) on the MRPC benchmark(Wang et al., 2019). Although MRPC is relatively small and noisy, this characteristic amplifies the impact of optimizer hyperparameters, making it easier to reveal differences in optimization behavior that may be less pronounced on larger, more stable datasets. At the same time, its modest size keeps the experiments computationally efficient while preserving representative fine-tuning dynamics of GLUE tasks. A preliminary sweep indicated that learning rates outside the range $[1 \times 10^{-5}, , 7 \times 10^{-5}]$ consistently led to poor accuracy; subsequent grids therefore focus on a narrower range centered at $2 \times 10^{-5}$.

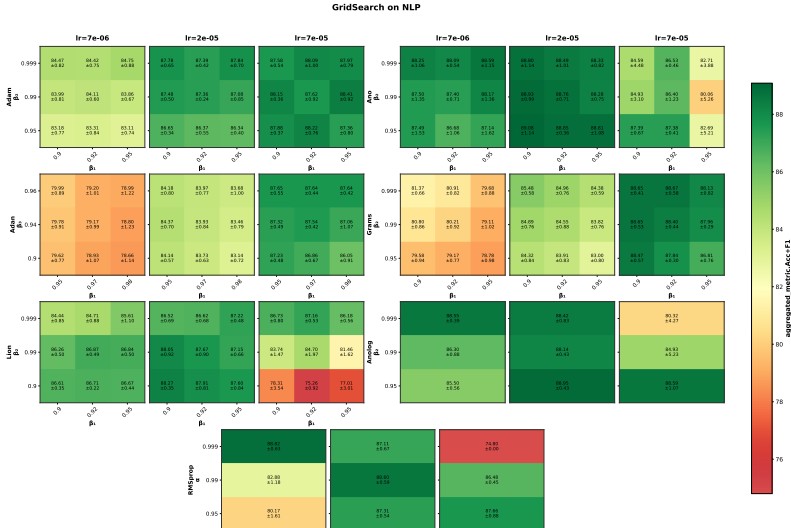

Figure 7: Grid-search results on MRPC used as an NLP proxy.

**Deep-RL proxy.** For Deep Reinforcement Learning, we train a SAC agent on the MuJoCo *HalfCheetah-v5* environment for 100k steps, given time constraints(Todorov et al., 2012; Haarnoja et al., 2018). This setup is primarily intended to reveal the impact of different hyperparameters, especially the momentum coefficients $\beta$, though we note that the shorter horizon may favor more aggressive learning rates. To address this limitation, the main text reports the best performance between default and tuned hyperparameters for each optimizer.

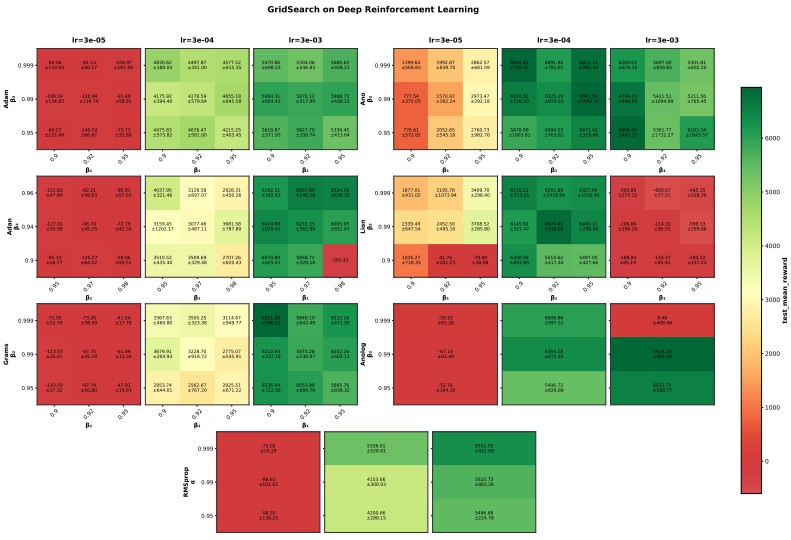

Figure 8: Grid-search on HalfCheetah 100k-steps training used as a DRL proxy.

# D CONVERGENCE PROOF FOR ANO

## D.1 ALGORITHMIC UPDATE

In line with standard analyses of adaptive methods (Reddi et al., 2018), we use delayed second-moment terms $v_{k-1}$ in the proof. This avoids the circular dependency between the update direction and the denominator, ensures that all quantities are measurable with respect to past information, and makes it possible to apply standard smoothness-based descent arguments. For each coordinate

$i \in [d]$ the optimizer maintains a first-order momentum $m_{k,i}$ and a second moment $v_{k,i}$ updated as below:

$$m_{k,i} = \beta_{1,k} m_{k-1,i} + (1 - \beta_{1,k}) g_{k,i}, \qquad v_{k,i} = \beta_2 v_{k-1,i} - (1 - \beta_2) \, \mathrm{sign}\!\left(v_{k-1,i} - g_{k,i}^2\right) g_{k,i}^2, \quad (1)$$

with $\beta_{1,k} = 1 - \frac{1}{\sqrt{k+1}}$ for $k \geq 1$; $\beta_2 \in [0.5, 1)$ and $m_{0,i} = v_{0,i} = 0$. The parameter vector is then updated by

$$x_{k+1,i} = x_{k,i} - \frac{\eta_k}{\sqrt{v_{k-1,i}} + \varepsilon} \, |g_{k,i}| \, \mathrm{sign}(m_{k,i}), \tag{2}$$

where $\varepsilon > 0$ is a fixed constant.

$$\eta_k = \frac{\eta}{(k+2)^{3/4}}, \qquad k = 0, 1, \dots \tag{3}$$

We write $\mathbb{E}_{k-1}[\cdot] := \mathbb{E}[\cdot \mid \mathcal{F}_{k-1}]$ for the conditional expectation given the filtration $\mathcal{F}_{k-1}$.

## D.2  STANDING ASSUMPTIONS

**Assumption 1** (Smoothness). *The objective function $f : \mathbb{R}^d \to \mathbb{R}$ is differentiable and $L$-smooth; that is, for all $x, y \in \mathbb{R}^d$,*
$$\|\nabla f(x) - \nabla f(y)\| \leq L\|x - y\|.$$

**Assumption 2** (Lower boundedness). *The function $f$ is bounded from below: there exists $f^\star > -\infty$ such that $f(x) \geq f^\star$ for all $x \in \mathbb{R}^d$.*

**Assumption 3** (Bounded gradients). *There exists a constant $G > 0$ such that $|\nabla_i f(x_k)| \leq G$ for all iterates $x_k$ and all coordinates $i$.*

**Assumption 4** (Unbiased stochastic gradients). *At each iteration $k$, we observe a stochastic gradient $g_k \in \mathbb{R}^d$ satisfying $\mathbb{E}[g_{k,i} \mid \mathcal{F}_{k-1}] = \nabla_i f(x_k)$ for all $i \in [d]$.*

**Assumption 5** (Bounded variance). *There exists $\sigma > 0$ such that for all $i \in [d]$ and $k \geq 1$,*
$$\mathbb{E}[(g_{k,i} - \nabla_i f(x_k))^2 \mid \mathcal{F}_{k-1}] \leq \sigma^2.$$

## D.3  PRELIMINARY LEMMA

**Local assumption.**  For this lemma only, we impose an additional assumption solely to simplify the analysis and to obtain a pointwise bound on $v_{k,i}$. Specifically, we assume $g_{k,i} \leq \tilde{G}$ for all $k, i$. This assumption is not used anywhere else in the paper and plays no role in the convergence results.

**Lemma 1** (Bounds on $v_k$). *Fix any coordinate $i \in [d]$ and assume $v_0 = 0$ and $\beta_2 \in [\frac{1}{2}, 1)$. Then for every $k \geq 0$,*
$$0 \;\leq\; v_{k,i} \;\leq\; \tilde{G}^2.$$

*Proof.* The update is
$$v_{k,i} = \beta_2 v_{k-1,i} - (1 - \beta_2) \, \mathrm{sign}(v_{k-1,i} - g_{k,i}^2) \, g_{k,i}^2.$$

**Upper bound.**  If $\mathrm{sign}(v_{k-1,i} - g_{k,i}^2) = -1$, then
$$v_{k,i} = \beta_2 v_{k-1,i} + (1 - \beta_2) g_{k,i}^2 \leq \tilde{G}^2$$

If $\mathrm{sign}(v_{k-1,i} - g_{k,i}^2) = 1$, then
$$v_{k,i} = \beta_2 v_{k-1,i} - (1 - \beta_2) g_{k,i}^2 \leq \beta_2 v_{k-1,i} \leq v_{k-1,i}.$$

Starting from $v_0 = 0$, induction gives $v_{k,i} \leq \tilde{G}^2$ for all $k$.

**Lower bound.**  If $\mathrm{sign}(v_{k-1,i} - g_{k,i}^2) = -1$, then $v_{k,i} \geq 0$ since it is a convex combination of nonnegative terms. If $\mathrm{sign}(v_{k-1,i} - g_{k,i}^2) = 1$, then
$$v_{k,i} = \beta_2 v_{k-1,i} - (1 - \beta_2) g_{k,i}^2 \geq (2\beta_2 - 1) g_{k,i}^2 \;\geq\; 0$$

because $\beta_2 \geq \frac{1}{2}$ and $g_{k,i}^2 \geq 0$. Thus $v_{k,i} \geq 0$ for all $k$. $\qquad\square$

### D.4 AUXILIARY QUANTITIES

Define the two per-iteration scalars

$$A_k := \sum_{i=1}^{d} \frac{\nabla_i f(x_k) \, |g_{k,i}| \, \text{sign}(m_{k,i})}{\sqrt{v_{k-1,i}} + \varepsilon}, \qquad B_k := \frac{L}{2} \sum_{i=1}^{d} \frac{g_{k,i}^2}{(\sqrt{v_{k-1,i}} + \varepsilon)^2}. \tag{4}$$

These two terms govern the decrease of the objective:

$$f(x_{k+1}) \leq f(x_k) - \eta_k A_k + \eta_k^2 B_k. \tag{5}$$

### D.5 LEMMA FOR PROBABILITY SIGN-MISMATCH

**Lemma 2** (Sign-Mismatch Probability for Ano). *Fix any coordinate $i \in [d]$, under Assumptions 1–5 and following update rules equation 2-equation 1, with $\beta_{1,k} = 1 - \frac{1}{\sqrt{k+1}}$ and $\eta_k = (k+2)^{-3/4}$. Then for every $k \geq 1$,*

$$\mathbb{P}(\text{sign}(m_{k,i}) \neq \text{sign}(\nabla_i f(x_k)) \leq \frac{C_m^2}{|\nabla_i f(x_k)|^2 \sqrt{k+1}}$$

*with $C_m := \sqrt{2(C_\Delta^2 + \sigma^2)}$ and $C_\Delta^2 = \frac{L^2 d(\sigma^2 + G^2)}{\varepsilon^2}$*

*Proof.* We fix a coordinate $i \in [d]$. Define the per-coordinate momentum error as

$$e_{k,i} := m_{k,i} - \nabla_i f(x_k).$$

Subtracting $\nabla_i f(x_k)$ from the update rule of $m_{k,i}$ yields:

$$e_{k,i} = \beta_{1,k} m_{k-1,i} + (1 - \beta_{1,k}) g_{k,i} - \nabla_i f(x_k)$$
$$= \beta_{1,k}(m_{k-1,i} - \nabla_i f(x_{k-1})) + \beta_{1,k}(\nabla_i f(x_{k-1}) - \nabla_i f(x_k)) + (1 - \beta_{1,k})(g_{k,i} - \nabla_i f(x_k))$$

Define $\Delta_{k,i} := \nabla_i f(x_{k-1}) - \nabla_i f(x_k)$ as the gradient variation, and $\xi_{k,i} := g_{k,i} - \nabla_i f(x_k)$ as the stochastic noise for coordinate $i$.

$$e_{k,i} = \beta_{1,k} e_{k-1,i} + \beta_{1,k} \Delta_{k,i} + (1 - \beta_{1,k}) \xi_{k,i},$$

Conditionally on $\mathcal{F}_{k-1}$, define $V_{k,i} := \mathbb{E}_{k-1}[e_{k,i}^2]$ :

$$V_{k,i} = \mathbb{E}_{k-1} \left[ (\beta_{1,k} e_{k-1,i} + \beta_{1,k} \Delta_{k,i} + (1 - \beta_{1,k}) \xi_{k,i})^2 \right].$$

Since $\mathbb{E}_{k-1}[\xi_{k,i}] = 0$ (Ass. 4), all mixed terms involving $\xi_{k,i}$ vanish after taking $\mathbb{E}_{k,i}$. Thus,

$$V_{k,i} = \mathbb{E}_{k-1} \left[ (\beta_{1,k} e_{k-1,i} + \beta_{1,k} \Delta_{k,i})^2 \right] + (1 - \beta_{1,k})^2 \mathbb{E}_{k-1}[\xi_{k,i}^2].$$

We now apply Young's inequality for scalars: $(a + b)^2 \leq (1 + \delta)a^2 + (1 + 1/\delta)b^2$ for any $\delta > 0$. We set $\delta = \frac{1}{\sqrt{k+1}}$ (this minimizes the resulting upper bound). Applying this gives:

$$\mathbb{E}_{k-1} \left[ (\beta_{1,k} e_{k-1,i} + \beta_{1,k} \Delta_{k,i})^2 \right] \leq \left( 1 + \frac{1}{\sqrt{k+1}} \right) \beta_{1,k}^2 \mathbb{E}_{k-1}[e_{k-1,i}^2] + (1 + \sqrt{k+1}) \beta_{1,k}^2 \mathbb{E}_{k-1}[\Delta_{k,i}^2].$$

Therefore,

$$V_{k,i} \leq \left( 1 + \frac{1}{\sqrt{k+1}} \right) \beta_{1,k}^2 V_{k-1,i} + (1 + \sqrt{k+1}) \beta_{1,k}^2 \mathbb{E}_{k-1}[\Delta_{k,i}^2] + (1 - \beta_{1,k})^2 \sigma^2.$$

To bound $\mathbb{E}_{k-1}[\Delta_{k,i}^2]$, we use the L-smoothness of $f$ (Assumption 1). We have:

$$|\Delta_{k,i}| = |\nabla_i f(x_{k-1}) - \nabla_i f(x_k)| \leq \|\nabla f(x_{k-1}) - \nabla f(x_k)\|_2 \leq L \|x_k - x_{k-1}\|_2.$$

Hence,

$$\mathbb{E}_{k-1}[\Delta_{k,i}^2] \leq L^2 \mathbb{E}_{k-1}[\|x_k - x_{k-1}\|_2^2].$$

Now we bound the step size:

$$\mathbb{E}_{k-1}[\|x_k - x_{k-1}\|_2^2] = \mathbb{E}_{k-1}\left[\sum_{j=1}^{d}\left(\frac{\eta_{k-1}}{\sqrt{v_{k-1,j}}+\varepsilon}|g_{k-1,j}|\right)^2\right]$$

$$\leq \eta_{k-1}^2 \sum_{j=1}^{d}\mathbb{E}_{k-1}\left[\frac{g_{k-1,j}^2}{\varepsilon^2}\right]$$

$$\leq \frac{\eta_{k-1}^2}{\varepsilon^2}\sum_{j=1}^{d}\mathbb{E}_{k-1}[g_{k-1,j}^2].$$

By Assumptions 3-5, we have $\mathbb{E}_{k-1}[g_{k-1,j}^2] \leq \sigma^2 + G^2$, where $\sigma^2$ is the variance bound and $G$ is an upper bound on the gradient norm. Thus,

$$\mathbb{E}_{k-1}[\|x_k - x_{k-1}\|_2^2] \leq \frac{d\eta_{k-1}^2(\sigma^2+G^2)}{\varepsilon^2}.$$

Let $C_\Delta^2 := \frac{L^2 d(\sigma^2+G^2)}{\varepsilon^2}$. Then, we obtain:

$$\mathbb{E}_{k-1}[\Delta_{k,i}^2] \leq C_\Delta^2 \eta_{k-1}^2.$$

Putting everything together, we get the recurrence:

$$V_{k,i} \leq \left(1 + \frac{1}{\sqrt{k+1}}\right)\beta_{1,k}^2 V_{k-1,i} + (1+\sqrt{k+1})\beta_{1,k}^2 C_\Delta^2 \eta_{k-1}^2 + (1-\beta_{1,k})^2\sigma^2.$$

Set $\beta_{1,k} = 1 - \frac{1}{\sqrt{k+1}}$ and $\eta_k = 1/(k+1)^{3/4}$, to simplify we denote $l = k+1$.

$$V_{k,i} \leq (1+\frac{1}{\sqrt{l}})(1-\frac{1}{\sqrt{l}})^2 V_{k-1,i} + (1+\sqrt{l})(1-\frac{1}{\sqrt{l}})^2 C_\Delta^2 \frac{1}{l^{3/2}} + \frac{\sigma^2}{k+1}$$

$$V_{k,i} \leq (1-\frac{1}{\sqrt{l}}-\frac{1}{l}+\frac{1}{l^{3/2}})V_{k-1,i} + \frac{1}{l}\left[(1+\frac{1}{\sqrt{l}})(1-\frac{1}{\sqrt{l}})^2 C_\Delta^2 + \sigma^2\right]$$

Hence $a_k := (1 - \frac{1}{\sqrt{l}} - \frac{1}{l} + \frac{1}{l^{3/2}})$ and $B_k := \frac{1}{l}\left[(1+\frac{1}{\sqrt{l}})(1-\frac{1}{\sqrt{l}})^2 C_\Delta^2 + \sigma^2\right]$, with $a_k \leq 1 - \frac{1}{\sqrt{l}}$ and $B_k = \mathcal{O}(l^{-1})$. So, we can express the inequality on this way

$$V_{k,i} \leq a_k V_{k-1,i} + B_k$$

Simplifying the coefficients. For $k \geq 1$,

$$a_k = 1 - \frac{1}{\sqrt{l}} - \frac{1}{l} + \frac{1}{l^{3/2}} \leq 1 - \frac{1}{\sqrt{l}}, \qquad (1+l^{-1/2})(1-l^{-1/2})^2 \leq 1,$$

so that

$$a_k \leq 1 - \frac{1}{\sqrt{l}}, \qquad B_k \leq \frac{C_B}{l}, \quad C_B := C_\Delta^2 + \sigma^2. \tag{6}$$

We want to simplify this bound by having a bound with the form $V_{k,i} \leq \mathcal{O}\left(\frac{1}{\sqrt{l}}\right)$, to proceed, we will prove by induction that $V_{k,i} = O(1/\sqrt{l})$. Specifically, we posit that there exists a constant $M$ such that for all sufficiently large $k$, $V_{k,i} \leq \frac{M}{\sqrt{l}}$.

**Base Case**   We can choose $M$ large enough such that the hypothesis holds for some initial $k_0 \geq 1$.

**Inductive Step** Assume that for some $k > k_0$, the hypothesis $V_{k-1} \leq \frac{M}{\sqrt{k-1}}$ holds. We must show that $V_{k,i} \leq \frac{M}{\sqrt{l}}$. From our simplified recurrence, we have:

$$V_{k,i} \leq \left(1 - \frac{1}{\sqrt{l}}\right) \frac{M}{\sqrt{l-1}} + \frac{C_B}{l}$$

The induction holds if we can prove:

$$\left(1 - \frac{1}{\sqrt{l}}\right) \frac{M}{\sqrt{l-1}} + \frac{C_B}{l} \leq \frac{M}{\sqrt{l}}$$

Rearranging the terms, this is equivalent to showing:

$$\frac{C_B}{l} \leq M \left(\frac{1}{\sqrt{l}} - \frac{1}{\sqrt{l-1}} + \frac{1}{\sqrt{l}\sqrt{l-1}}\right)$$

To analyze the right-hand side (RHS) for large $k$, we use a Taylor expansion for the term $(l-1)^{-1/2}$:

$$\frac{1}{\sqrt{l-1}} = (l-1)^{-1/2} = l^{-1/2} \left(1 - \frac{1}{l}\right)^{-1/2}$$

Using the expansion $(1-x)^{-1/2} = 1 + \frac{x}{2} + O(x^2)$, we get:

$$\frac{1}{\sqrt{l-1}} = \frac{1}{\sqrt{l}} \left(1 + \frac{1}{2l} + O\left(\frac{1}{l^2}\right)\right) = \frac{1}{\sqrt{l}} + \frac{1}{2l^{3/2}} + O\left(\frac{1}{l^{5/2}}\right)$$

Substituting this into the parenthesis on the RHS of our inequality, the term becomes:

$$\frac{1}{\sqrt{l}} - \left(\frac{1}{\sqrt{l}} + \frac{1}{2l^{3/2}}\right) + \frac{1}{l\sqrt{1-1/l}} + O\left(\frac{1}{l^{5/2}}\right)$$

$$= -\frac{1}{2l^{3/2}} + \frac{1}{l}\left(1 + \frac{1}{2l} + O\left(\frac{1}{l^2}\right)\right) + O\left(\frac{1}{l^{5/2}}\right)$$

$$= \frac{1}{l} - \frac{1}{2l^{3/2}} + O\left(\frac{1}{l^2}\right)$$

The full inequality we need to satisfy is therefore:

$$\frac{C_B}{l} \leq M \left(\frac{1}{l} - \frac{1}{2l^{3/2}} + O\left(\frac{1}{l^2}\right)\right)$$

Multiplying through by $l$, the condition becomes:

$$C_B \leq M \left(1 - \frac{1}{2\sqrt{l}} + O\left(\frac{1}{l}\right)\right)$$

This inequality shows why the induction works. For any choice of constant $M > C_B$, we can find a sufficiently large $k_0$ such that for all $k \geq k_0$, the inequality holds. This completes the induction, establishing that $V_{k,i} \leq \frac{M}{\sqrt{l}}$. So, for a sufficiently large $k_0$ such that for all $k \geq k_0$, we have :

$$V_{k,i} \leq \frac{2C_B}{\sqrt{l}}$$

$$\mathbb{E}_{k-1}[(m_{k,i} - \nabla_i f(x_k))^2] \leq \frac{2(C_\Delta^2 + \sigma^2)}{\sqrt{l}}$$

$$\mathbb{E}_{k-1}[|m_{k,i} - \nabla_i f(x_k)|^2] \leq \frac{2(C_\Delta^2 + \sigma^2)}{\sqrt{l}}$$

We recall that $l = k + 1$, so

$$\mathbb{E}_{k-1}[|m_{k,i} - \nabla_i f(x_k)|^2] \leq \frac{2(C_\Delta^2 + \sigma^2)}{\sqrt{k+1}}$$

**From moment bound to probability bound** We bound the probability of a momentum–gradient sign mismatch. If $\text{sign}(m_{k,i}) \neq \text{sign}(\nabla_i f(x_k))$ and $\nabla_i f(x_k) \neq 0$, then $|m_{k,i} - \nabla_i f(x_k)| \geq |\nabla_i f(x_k)|$. Hence, for any $k \geq 1$,

$$\mathbb{P}_{k-1}\big(\text{sign}(m_{k,i}) \neq \text{sign}(\nabla_i f(x_k))\big) \leq \mathbb{P}\big(|m_{k,i} - \nabla_i f(x_k)| \geq |\nabla_i f(x_k)|\big)$$

We apply Chebyshev's inequality to the right-hand side:

$$\mathbb{P}_{k-1}\big(|m_{k,i} - \nabla_i f(x_k)| \geq |\nabla_i f(x_k)|\big) \leq \frac{\mathbb{E}_{k-1}\big[|m_{k,i} - \nabla_i f(x_k)|^2\big]}{|\nabla_i f(x_k)|^2}$$

Using the previously established second-moment bound, $\mathbb{E}_{k-1}[|m_{k,i} - \nabla_i f(x_k)|^2] \leq \frac{2(C_\Delta^2 + \sigma^2)}{\sqrt{k+1}}$ :

$$\mathbb{P}_{k-1}\big(\text{sign}(m_{k,i}) \neq \text{sign}(\nabla_i f(x_k))\big) \leq \frac{C_m^2}{|\nabla_i f(x_k)|^2 \sqrt{k+1}}$$

with $C_m = \sqrt{2(C_\Delta^2 + \sigma^2)}$ $\qquad\qquad\qquad\qquad\qquad\qquad\qquad\qquad\qquad\qquad\qquad$ $\square$

## D.6 BOUND ON A

**Lemma 3** (Lower bound on the expected update magnitude). *Assume all the condition (cf D.2. Recall*

$$A_k = \sum_{i=1}^d \frac{\nabla_i f(x_k) \, |g_{k,i}| \, \text{sign}(m_{k,i})}{\sqrt{v_{k-1,i}} + \varepsilon}, \qquad \mathbb{E}_{k-1}[\cdot] = \mathbb{E}[\cdot \mid \mathcal{F}_{k-1}].$$

*Let*

$$C_m = \sqrt{2(C_\Delta^2 + \sigma^2)}, \quad C_v = \frac{2d\sqrt{\sigma^2 + G^2}}{\varepsilon}$$

*Then, for every iteration $k \geq k_0$,*

$$\mathbb{E}[A_k] \geq \frac{\mathbb{E}[\|\nabla f(x_k)\|_2^2]}{\tilde{G} + \varepsilon} - \frac{C_v C_m}{(k+1)^{1/4}},$$

*Proof.* We begin by recalling the definition of $A_k$ and factoring out constants that do not depend on $g_{k,i}$:

$$\mathbb{E}_{k-1}[A_k] = \sum_{i=1}^d \frac{\nabla_i f(x_k)}{\sqrt{v_{k-1,i}} + \varepsilon} \cdot \mathbb{E}_{k-1}\big[|g_{k,i}| \, \text{sign}(m_{k,i})\big].$$

Our goal is to lower bound the term $\mathbb{E}_{k-1}\big[|g_{k,i}| \, \text{sign}(m_{k,i})\big]$.

We first expand this term using the identity:

$$\text{sign}(m_{k,i}) = \text{sign}(\nabla_i f(x_k)) \cdot (1 - 2 \cdot \mathbb{I}\,[\text{sign}(\nabla_i f(x_k)) \neq \text{sign}(m_{k,i})])\,,$$

Let $\chi_{k,i} := \mathbb{I}\,[\text{sign}(\nabla_i f(x_k)) \neq \text{sign}(m_{k,i})]$. , we get:

$$\mathbb{E}_{k-1}\big[|g_{k,i}| \, \text{sign}(m_{k,i})\big] = \text{sign}(\nabla_i f(x_k)) \cdot \mathbb{E}_{k-1}\,[|g_{k,i}|]$$
$$- 2 \cdot \mathbb{E}_{k-1}\,[|g_{k,i}| \cdot \text{sign}(\nabla_i f(x_k)) \cdot \chi_{k,i}]$$

where the second line follows from linearity of expectation.

We now bound the first term using Jensen

$$\text{sign}(\nabla_i f(x_k)) \cdot \mathbb{E}_{k-1}[|g_{k,i}|] \geq \text{sign}(\nabla_i f(x_k)) \cdot |\mathbb{E}_{k-1}[g_{k,i}]| = \nabla_i f(x_k),$$

where we used the assumption that $\mathbb{E}_{k-1}[g_{k,i}] = \nabla_i f(x_k)$.

We bound the second term by combining $\text{sign}(\nabla_i f(x_k)) \leq 1$ and Cauchy-Schwarz:

$$\mathbb{E}_{k-1}\,[|g_{k,i}| \cdot \text{sign}(\nabla_i f(x_k)) \cdot \chi_{k,i}] \leq \sqrt{\mathbb{E}_{k-1}\left[g_{k,i}^2\right]} \cdot \sqrt{\mathbb{E}_{k-1}\left[\chi_{k,i}^2\right]}$$

By the variance definition, we know that $\mathrm{Var}(X) = \mathbb{E}[X^2] - (\mathbb{E}[X])^2$, so we have

$$\mathbb{E}_{k-1}\left[g_{k,i}^2\right] = \mathrm{Var}_{k-1}(g_{k,i}) + (\mathbb{E}_{k-1}[g_{k,i}])^2$$

By combining Assumptions (4, 3 and 5), and $\chi_{k,i}^2 = \chi_{k,i}$, we got :

$$\mathbb{E}_{k-1}\left[|g_{k,i}| \cdot \mathrm{sign}(\nabla_i f(x_k)) \cdot \chi_{k,i}\right] \leq \sqrt{\sigma^2 + G^2} \cdot \sqrt{\mathbb{P}_{k-1}\left(\chi_{k,i}\right)}$$

By using lemma 2,

$$\mathbb{E}_{k-1}\left[|g_{k,i}| \cdot \mathrm{sign}(\nabla_i f(x_k)) \cdot \chi_{k,i}\right] \leq \sqrt{\sigma^2 + G^2} \cdot \frac{C_m}{|\nabla_i f(x_k)|(k+1)^{1/4}}$$

We have so :

$$\mathbb{E}_{k-1}\left[|g_{k,i}| \, \mathrm{sign}(m_{k,i})\right] \geq \nabla_i f(x_k) - 2 \cdot \sqrt{\sigma^2 + G^2} \cdot \frac{C_m}{|\nabla_i f(x_k)|(k+1)^{1/4}}$$

In our main equation, we have then

$$\mathbb{E}_{k-1}[A_k] \geq \sum_{i=1}^{d} \frac{\nabla_i f(x_k)}{\sqrt{v_{k-1,i}} + \varepsilon} \cdot \left(\nabla_i f(x_k) - 2 \cdot \sqrt{\sigma^2 + G^2} \cdot \frac{C_m}{|\nabla_i f(x_k)|(k+1)^{1/4}}\right)$$

$$\mathbb{E}_{k-1}[A_k] \geq \sum_{i=1}^{d} \frac{(\nabla_i f(x_k))^2}{\sqrt{v_{k-1,i}} + \varepsilon} - \frac{1}{(k+1)^{1/4}} \cdot 2\sqrt{\sigma^2 + G^2} \, C_m \sum_{i=1}^{d} \frac{1}{\sqrt{v_{k-1,i}} + \varepsilon}$$

Using lemma 1, we got $0 \leq v_{k-1,i} \leq \tilde{G}^2$, we deduce:

$$\mathbb{E}_{k-1}[A_k] \geq \frac{\|\nabla f(x_k)\|_2^2}{\tilde{G} + \varepsilon} - \frac{1}{(k+1)^{1/4}} \cdot \frac{2d\sqrt{\sigma^2 + G^2} \, C_m}{\varepsilon}$$

Finally, letting $C_v = \frac{2d\sqrt{\sigma^2 + G^2}}{\varepsilon}$, we can write:

$$\mathbb{E}_{k-1}[A_k] \geq \frac{\|\nabla f(x_k)\|_2^2}{\tilde{G} + \varepsilon} - \frac{C_v C_m}{(k+1)^{1/4}},$$

By the total expectation law

$$\mathbb{E}[A_k] \geq \frac{\mathbb{E}[\|\nabla f(x_k)\|_2^2]}{\tilde{G} + \varepsilon} - \frac{C_v C_m}{(k+1)^{1/4}},$$

which concludes the proof. $\qquad\square$

### D.7 BOUND ON B

**Lemma 4.** *Assume the standing hypotheses of the paper hold, in particular (Assumptions 3-5) $\mathbb{E}_{k-1}[g_{k,i}^2] \leq G^2 + \sigma^2$, for all time-steps $k$ and coordinates $i$, and let $\varepsilon > 0$. Then, for every iteration $k \geq 1$,*

$$\mathbb{E}[B_k] \leq \frac{L\,d\,(G^2 + \sigma^2)}{2\,\varepsilon^2}$$

*Proof.* Because $v_{k-1,i} \geq 0$ and $\varepsilon > 0$, we have

$$(\sqrt{v_{k-1,i}} + \varepsilon)^2 \geq \varepsilon^2.$$

Together with the bound $\mathbb{E}_{k-1}[g_{k,i}^2] \leq G^2 + \sigma^2$, this implies

$$\mathbb{E}_{k-1}\left[\frac{g_{k,i}^2}{(\sqrt{v_{k-1,i}} + \varepsilon)^2}\right] \leq \frac{G^2 + \sigma^2}{\varepsilon^2} \quad \text{for all } i.$$

Summing over $i = 1, \ldots, d$ and factoring out $L/2$ yields

$$\mathbb{E}_{k-1}[B_k] \leq \frac{L}{2}\,d\,\frac{(G^2 + \sigma^2)}{\varepsilon^2} = \frac{L\,d\,(G^2 + \sigma^2)}{2\,\varepsilon^2},$$

By the total expectation law

$$\mathbb{E}[B_k] \leq \frac{L\,d\,(G^2 + \sigma^2)}{2\,\varepsilon^2}$$

which completes the proof. $\qquad\square$

## D.8  MAIN RESULT

**Theorem 1** (Convergence to a stationary point). *Let Assumption D.2 hold and set the learning rate as in equation 3. Then for any horizon $K \geq 1$,*

$$\min_{0 \leq k < K} \mathbb{E}\big[\|\nabla f(x_k)\|_2^2\big] \leq \mathcal{O}\left(\frac{\log K}{K^{1/4}}\right).$$

*Proof.* The proof starts from the descent guarantee provided by the $L$-smoothness of $f(x)$, as stated in equation 5:

$$f(x_{k+1}) \leq f(x_k) - \eta_k A_k + \eta_k^2 B_k.$$

Taking $\mathbb{E}_{k-1}$, we get:

$$\mathbb{E}[f(x_{k+1})] \leq \mathbb{E}[f(x_k)] - \eta_k \mathbb{E}[A_k] + \eta_k^2 \mathbb{E}[B_k].$$

We now bound the terms $\mathbb{E}[A_k]$ and $\mathbb{E}[B_k]$ using the provided lemmas.

1. **Bounding** $\mathbb{E}[A_k]$: From Lemma 3, we have:

$$\mathbb{E}[A_k] \geq \frac{1}{\sqrt{2}\,G + \varepsilon} \mathbb{E}[\|\nabla f(x_k)\|_2^2] - \frac{C_m C_v}{(k+1)^{\frac{1}{4}}}$$

2. **Bounding** $\mathbb{E}[B_k]$: From Lemma 4, $B_k$ is uniformly bounded. Define $C_b := \frac{L\,d\,(G^2 + \sigma^2)}{2\,\varepsilon^2}$, then:

$$\mathbb{E}[B_k] \leq C_b.$$

Substituting into the main inequality:

$$\mathbb{E}[f(x_{k+1})] \leq \mathbb{E}[f(x_k)] - \frac{\eta_k}{\sqrt{2}\,G + \varepsilon} \mathbb{E}\big[\|\nabla f(x_k)\|_2^2\big] + \eta_k \frac{C_m C_v}{k^{1/4}} + \eta_k^2 C_b.$$

Let $\eta_k = \frac{\eta}{(k+2)^{3/4}} \leq \frac{\eta}{(k+1)^{3/4}}$. Then:

$$\mathbb{E}[f(x_{k+1})] \leq \mathbb{E}[f(x_k)] - \frac{\eta}{(k+1)^{3/4}(\sqrt{2}\,G + \varepsilon)} \mathbb{E}\big[\|\nabla f(x_k)\|_2^2\big] + \frac{C_m C_v \eta}{k+1} + \frac{\eta^2\,C_b}{(k+1)^{3/2}}$$

Rewriting:

$$\frac{\eta}{(k+1)^{3/4}(\sqrt{2}\,G + \varepsilon)} \mathbb{E}\big[\|\nabla f(x_k)\|_2^2\big] \leq \mathbb{E}[f(x_k)] - \mathbb{E}[f(x_{k+1})] + \frac{C_m C_v \eta}{k+1} + \frac{\eta^2\,C_b}{(k+1)^{3/2}}$$

Summing from $k = 0$ to $K - 1$:

$$\sum_{k=0}^{K-1} \frac{\eta}{(k+1)^{3/4}(\sqrt{2}\,G + \varepsilon)} \mathbb{E}\big[\|\nabla f(x_k)\|_2^2\big] \leq f(x_0) - f^\star + C_m C_v \eta \sum_{k=0}^{K-1} \frac{1}{k+1} + \eta^2\,C_b \sum_{k=0}^{K-1} \frac{1}{(k+1)^{3/2}}$$

The harmonic sum satisfies:

$$\sum_{k=0}^{K-1} \frac{1}{k+1} \leq 1 + \log K.$$

The second term is a convergent $p$-series with exponent $p = 3/2 > 1$; hence it converges to a finite limit as $K \to \infty$. Therefore, the partial sum can be bounded by the value of the full series:

$$\eta^2\,C_b \sum_{k=0}^{K-1} (k+1)^{-3/2} \; \leq \; \eta^2\,C_b \sum_{k=0}^{\infty} k^{-3/2} \; = \; \eta^2\,C_b\,\zeta\big(\tfrac{3}{2}\big) \; =: C_R,$$

where $\zeta(\cdot)$ denotes the Riemann zeta function. We thus define the constant $C_R := \eta^2 C_b \zeta\big(\tfrac{3}{2}\big)$.

By combining these terms, we have for the main equation:

$$\sum_{k=0}^{K-1} \frac{1}{(k+1)^{3/4}} \mathbb{E}\big[\|\nabla f(x_k)\|_2^2\big] \leq \frac{1}{C_{LHS}} \left[f(x_0) - f^\star + C_{sum}(1 + \log K) + C_R\right]$$

with $C_{sum} = C_m C_v \eta$ and $C_{LHS} = \frac{\eta}{\tilde{G}+\epsilon}$.

The left-hand side can be lower-bounded as follows:

$$\sum_{k=1}^{K-1} \frac{1}{(k+1)^{3/4}} \mathbb{E}\big[\|\nabla f(x_k)\|_2^2\big] \geq \min_{0 \leq k < K} \mathbb{E}\big[\|\nabla f(x_k)\|_2^2\big] \sum_{k=1}^{K-1} \frac{1}{(k+1)^{3/4}}$$

Now, let's find a lower bound for the sum. We can approximate it with an integral:

$$\sum_{k=0}^{K-1} \frac{1}{(k+1)^{3/4}} = \sum_{j=1}^{K} \frac{1}{j^{3/4}} \geq \int_1^{K+1} \frac{1}{x^{3/4}} dx = \left[4x^{1/4}\right]_1^{K+1} = 4\left((K+1)^{1/4} - 1\right)$$

For a large $K$, this sum is of the order $\mathcal{O}(K^{1/4})$.

Substituting this back into our main inequality, we get:

$$4\left((K+1)^{1/4} - 1\right) \min_{0 \leq k < K} \mathbb{E}\big[\|\nabla f(x_k)\|_2^2\big] \leq \frac{1}{C_{LHS}} \left[f(x_0) - f^\star + C_{sum}(1 + \log K) + C_R\right]$$

Now, we can isolate the minimum of the expected squared norm of the gradient:

$$\min_{0 \leq k < K} \mathbb{E}\big[\|\nabla f(x_k)\|_2^2\big] \leq \frac{f(x_0) - f^\star + C_{sum}(1 + \log K) + C_R}{4C_{LHS}\left((K+1)^{1/4} - 1\right)}$$

As $K \to \infty$, the dominant terms are $\log K$ in the numerator and $K^{1/4}$ in the denominator. Therefore, we can write the convergence rate as:

$$\min_{0 \leq k < K} \mathbb{E}\big[\|\nabla f(x_k)\|_2^2\big] = \mathcal{O}\left(\frac{\log K}{K^{1/4}}\right) = \tilde{\mathcal{O}}\left(\frac{1}{K^{1/4}}\right)$$

**Remark 1** (On the choice of hyperparameters). *The convergence analysis required carefully tuned hyperparameters. In particular, we employed a progressively increasing momentum coefficient, $\beta_{1,k} = 1 - 1/\sqrt{k}$, along with a relatively aggressive learning rate schedule. This combination was essential to control key residual terms throughout the proof and to ensure overall convergence. While more conservative choices failed to yield meaningful bounds, the schedule used here proved sufficient for our theoretical guarantees.*

**Remark 2** (On the convergence rate). *The convergence rate $\mathcal{O}(\log K/K^{1/4})$ for $\min_k \mathbb{E}[\|\nabla f(x_k)\|^2]$ arises intrinsically from the sign-based nature of the update rule. Unlike gradient-magnitude-based methods, our approach relies solely on the sign of momentum terms, which demands high directional accuracy. Ensuring this accuracy requires a tight control over the momentum error variance, which in turn necessitates a fast-decaying learning rate, $\eta_k = \mathcal{O}(k^{-3/4})$. This schedule guarantees reliable update directions but slows the overall convergence, as the learning rate bounds the algorithm's progress. Hence, the rate reflects a fundamental trade-off: stability of sign-based directions versus the speed of descent.*

$\square$

# E ADDITIONAL RESULTS

## E.1 FULL RESULTS ON ANALYSIS PARTS

## E.2 FULL NORMALIZED SCORE FOR PPO

# F HYPERPARAMETERS SETTINGS

## F.1 OPTIMIZERS HYPERPARAMETERS

| Optimizer | $\sigma = 0$ | 0.01 | 0.05 | 0.10 | 0.20 |
|---|---|---|---|---|---|
| Ano | 82.10±0.20 | 78.71±0.20 | 70.88±0.34 | 65.93±0.33 | 59.54±0.66 |
| Adam | 80.67±0.37 | 75.97±0.27 | 66.86±0.52 | 60.83±0.54 | 52.46±0.93 |
| Lion | 81.04±0.30 | 77.80±0.19 | 69.62±0.28 | 64.02±0.77 | 56.82±0.78 |
| Grams | 71.34±0.33 | 77.90±0.02 | 70.57±0.23 | 65.47±0.32 | 58.80±0.57 |

Table 9: CIFAR-10 test accuracy (%) with 95% confidence intervals.

| Optimizers | HalfCheetah-v5 Score±IC | Norm | Ant-v5 Score±IC | Norm | Humanoid-v5 Score±IC | Norm | Walker2d-v5 Score±IC | Norm | Hopper-v5 Score±IC | Norm | Avg. Norm |
|---|---|---|---|---|---|---|---|---|---|---|---|
| *Default* | | | | | | | | | | | |
| Adam | 10549.48 ± 721.55 | 97.10 | 4336.64 ± 698.72 | 82.05 | 5357.14 ± 211.97 | 99.29 | 4462.51 ± 588.77 | 85.36 | 3164.71 ± 600.48 | 89.52 | 90.66 |
| RMSprop | 10506.23 ± 852.19 | 96.71 | 4234.37 ± 763.65 | 80.11 | 5395.51 ± 126.80 | 100.00 | 4160.06 ± 480.62 | 79.57 | 2973.86 ± 571.05 | 84.12 | 88.10 |
| Adan | 7805.20 ± 1154.02 | 71.84 | 2985.19 ± 1018.79 | 56.48 | 5080.74 ± 305.26 | 94.17 | 4092.13 ± 379.92 | 78.28 | 3222.62 ± 235.25 | 91.16 | 78.38 |
| Lion | 9527.96 ± 805.42 | 87.70 | 4948.26 ± 243.05 | 93.62 | 98.22 ± 32.33 | 1.82 | 4612.63 ± 367.77 | 88.23 | 3087.27 ± 628.06 | 87.33 | 71.74 |
| Grams | 6782.60 ± 715.12 | 62.43 | 3207.30 ± 531.06 | 60.68 | 5104.10 ± 692.14 | 94.60 | 3656.66 ± 658.82 | 69.95 | 1475.34 ± 927.22 | 41.73 | 65.88 |
| Ano (Ours) | 10864.09 ± 1052.24 | 100.00 | 5285.44 ± 729.86 | 100.00 | 5255.62 ± 815.92 | 97.41 | 5227.86 ± 436.49 | 100.00 | 3535.32 ± 780.96 | 100.00 | 99.48 |
| Anolog (Ours) | 10557.05 ± 560.70 | 97.17 | 5089.12 ± 522.94 | 96.29 | 5242.78 ± 173.98 | 97.17 | 4606.02 ± 478.36 | 88.11 | 3314.12 ± 539.95 | 93.74 | 94.50 |
| *Tuned* | | | | | | | | | | | |
| Adam | 8243.01 ± 2750.47 | 69.70 | 5050.53 ± 471.12 | 90.82 | 5224.24 ± 339.87 | 100.00 | 4429.62 ± 668.97 | 90.83 | 2968.40 ± 696.52 | 82.62 | 86.79 |
| RMSprop | 10096.62 ± 2379.00 | 85.37 | 3509.99 ± 827.41 | 63.12 | 64.97 ± 42.44 | 1.24 | 4583.19 ± 969.52 | 93.98 | 2031.80 ± 771.10 | 56.55 | 60.05 |
| Adan | 10822.40 ± 475.75 | 91.51 | 5239.69 ± 270.96 | 94.22 | 4792.62 ± 904.44 | 91.74 | 4686.83 ± 502.28 | 96.11 | 3514.42 ± 143.57 | 97.82 | 94.28 |
| Lion | 10482.06 ± 1018.86 | 88.63 | 4848.41 ± 821.79 | 87.18 | 1349.15 ± 1322.56 | 25.82 | 4876.76 ± 253.22 | 100.00 | 3592.87 ± 70.26 | 100.00 | 80.33 |
| Grams | 10533.70 ± 866.69 | 89.07 | 4607.59 ± 505.08 | 82.85 | 5147.04 ± 487.55 | 98.52 | 4644.45 ± 498.08 | 95.24 | 3147.82 ± 605.03 | 87.61 | 90.66 |
| Ano (Ours) | 11826.22 ± 700.46 | 100.00 | 5561.17 ± 400.26 | 100.00 | 5158.30 ± 313.97 | 98.74 | 4804.34 ± 359.02 | 98.51 | 3226.05 ± 504.58 | 89.79 | 97.41 |
| Anolog (Ours) | 11198.28 ± 771.94 | 94.69 | 5095.64 ± 722.28 | 91.63 | 3137.03 ± 1335.43 | 60.05 | 4563.86 ± 834.12 | 93.58 | 3321.91 ± 472.93 | 92.46 | 86.48 |
| *Best Version* | | | | | | | | | | | |
| Adam | 10549.48 ± 721.55 | 97.10 | 4336.64 ± 698.72 | 82.05 | 5357.14 ± 211.97 | 99.29 | 4462.51 ± 588.77 | 85.36 | 3164.71 ± 600.48 | 88.08 | 90.38 |
| RMSprop | 10506.23 ± 852.19 | 96.71 | 4234.37 ± 763.65 | 80.11 | 5395.51 ± 126.80 | 100.00 | 4160.06 ± 480.62 | 79.57 | 2973.86 ± 571.05 | 82.77 | 87.83 |
| Adan | 10822.40 ± 475.75 | 99.62 | 5239.69 ± 270.96 | 99.13 | 4792.62 ± 904.44 | 88.83 | 4686.83 ± 502.28 | 89.65 | 3514.42 ± 143.57 | 97.82 | 95.01 |
| Lion | 10482.06 ± 1018.86 | 96.48 | 4848.41 ± 821.79 | 91.73 | 1349.15 ± 1322.56 | 25.01 | 4876.76 ± 253.22 | 93.28 | 3592.87 ± 70.26 | 100.00 | 81.30 |
| Grams | 10533.70 ± 866.69 | 96.96 | 4607.59 ± 505.08 | 87.18 | 5147.04 ± 487.55 | 95.39 | 4644.45 ± 498.08 | 88.84 | 3147.82 ± 605.03 | 87.61 | 91.20 |
| Ano (Ours) | 10864.09 ± 1052.24 | 100.00 | 5285.44 ± 729.86 | 100.00 | 5255.62 ± 815.92 | 97.41 | 5227.86 ± 436.49 | 97.41 | 3535.32 ± 780.96 | 98.40 | 99.16 |
| Anolog (Ours) | 10557.05 ± 560.70 | 97.17 | 5089.12 ± 522.94 | 96.29 | 5242.78 ± 173.98 | 97.17 | 4606.02 ± 478.36 | 97.17 | 3314.12 ± 539.95 | 92.24 | 94.20 |

Table 10: Comparison of the average performance (± CI95%) and normalized scores of different optimizers across MuJoCo environments.

Table 12: Optimizers hyperparameter settings used in all experiments.

| Model | Optimizer | $\beta_1$ | $\beta_2$ | $\beta_3$ | $lr$ | $\lambda$ |
|---|---|---|---|---|---|---|
| *Noise Robustness Analysis* | | | | | | |
| CNN | AdamW | 0.9 | 0.999 | – | 1e-3 | – |
| | Lion | 0.9 | 0.99 | – | 1e-4 | – |
| | Ano | 0.92 | 0.99 | – | 1e-4 | – |
| | Grams | 0.9 | 0.999 | – | 1e-3 | – |
| *Non-Stationnary Analysis* | | | | | | |
| | Adam | 0.9 | 0.999 | – | 1e-3 | – |
| | Lion | 0.9 | 0.9 | – | 1e-3 | – |
| | Ano | 0.9 | 0.999 | – | 1e-3 | – |
| | Grams | 0.9 | 0.999 | – | 1e-3 | – |
| *Computer Vision (CIFAR-100)* | | | | | | |
| *Baseline* | | | | | | |
| ResNet-34 | AdamW | 0.9 | 0.999 | – | 1e-3 | 1e-2 |
| | Adan | 0.98 | 0.92 | 0.99 | 1e-3 | 1e-2 |
| | Ano | 0.92 | 0.99 | – | 1e-3 | 1e-2 |
| | Lion | 0.9 | 0.99 | – | 1e-3 | 1e-2 |
| | Grams | 0.9 | 0.999 | – | 1e-3 | 1e-2 |
| *Tuned* | | | | | | |
| ResNet-34 | AdamW | 0.9 | 0.99 | – | 1e-3 | 1e-2 |
| | Adan | 0.95 | 0.92 | 0.9 | 1e-3 | 1e-2 |
| | Ano | 0.95 | 0.95 | – | 1e-4 | 1e-2 |
| | Lion | 0.92 | 0.99 | – | 1e-4 | 1e-2 |
| | Grams | 0.9 | 0.99 | – | 1e-3 | 1e-2 |
| *Natural Language Processing (GLUE)* | | | | | | |
| *Baseline* | | | | | | |

**Table 12** (continued)

| Model | Optimizer | $\beta_1$ | $\beta_2$ | $\beta_3$ | $lr$ | $\lambda$ |
|---|---|---|---|---|---|---|
| BERT (base) | AdamW | 0.9 | 0.999 | – | 2e-5 | 1e-2 |
| | Adan | 0.98 | 0.92 | 0.99 | 2e-5 | 1e-2 |
| | Ano | 0.92 | 0.99 | – | 2e-5 | 1e-2 |
| | Lion | 0.9 | 0.99 | – | 2e-5 | 1e-2 |
| | Grams | 0.9 | 0.999 | – | 2e-5 | 1e-2 |
| | Anolog | – | 0.999 | – | 2e-5 | 1e-2 |
| *Tuned* | | | | | | |
| BERT (base) | AdamW | 0.95 | 0.99 | – | 7e-5 | 1e-2 |
| | Adan | 0.95 | 0.92 | 0.96 | 7e-5 | 1e-2 |
| | Ano | 0.9 | 0.95 | – | 2e-5 | 1e-2 |
| | Lion | 0.9 | 0.9 | – | 7e-6 | 1e-2 |
| | Grams | 0.92 | 0.999 | – | 7e-5 | 1e-2 |
| | Anolog | – | 0.95 | – | 2e-5 | 1e-2 |
| *Deep Reinforcement Learning (MuJoCo & Atari)* | | | | | | |
| *Baseline* | | | | | | |
| SAC/PPO | Adam | 0.9 | 0.999 | – | 3e-4 | – |
| | Adan | 0.98 | 0.92 | 0.99 | 3e-4 | – |
| | Ano | 0.92 | 0.99 | – | 3e-4 | – |
| | Lion | 0.9 | 0.99 | – | 3e-4 | – |
| | Grams | 0.9 | 0.999 | – | 3e-4 | – |
| | RMSprop | – | 0.99 | – | 3e-4 | – |
| | Anolog | – | 0.999 | – | 3e-4 | – |
| *Tuned* | | | | | | |
| SAC/PPO | Adam | 0.9 | 0.99 | – | 3e-3 | – |
| | Adan | 0.97 | 0.92 | 0.96 | 3e-3 | – |
| | Ano | 0.95 | 0.99 | – | 3e-4 | – |
| | Lion | 0.92 | 0.99 | – | 3e-4 | – |
| | Grams | 0.9 | 0.999 | – | 3e-3 | – |
| | RMSprop | – | 0.999 | – | 3e-3 | – |
| | Anolog | – | 0.99 | – | 3e-3 | – |
| *Best Version (SAC)* | | | | | | |
| SAC | Adam | 0.9 | 0.999 | – | 3e-4 | – |
| | Adan | 0.97 | 0.92 | 0.96 | 3e-3 | – |
| | Ano | 0.92 | 0.99 | – | 3e-4 | – |
| | Lion | 0.92 | 0.99 | – | 3e-4 | – |
| | Grams | 0.9 | 0.999 | – | 3e-3 | – |
| | RMSprop | – | 0.99 | – | 3e-4 | – |
| | Anolog | – | 0.999 | – | 3e-4 | – |
| *Best Version (PPO)* | | | | | | |
| PPO | Adam | 0.9 | 0.999 | – | 3e-4 | – |
| | Adan | 0.97 | 0.92 | 0.96 | 3e-3 | – |
| | Ano | 0.95 | 0.99 | – | 3e-4 | – |
| | Lion | 0.92 | 0.99 | – | 3e-4 | – |
| | Grams | 0.9 | 0.999 | – | 3e-3 | – |
| | RMSprop | – | 0.99 | – | 3e-4 | – |
| | Anolog | – | 0.999 | – | 3e-4 | – |

| Optimizers | BattleZone-v5 | | DoubleDunk-v5 | | NameThisGame-v5 | | Phoenix-v5 | | Qbert-v5 | | Avg. Norm |
|---|---|---|---|---|---|---|---|---|---|---|---|
| | Score±IC | Norm | Score±IC | Norm | Score±IC | Norm | Score±IC | Norm | Score±IC | Norm | |
| *Default* | | | | | | | | | | | |
| Adam | 7615.00 ± 1299.89 | 94.07 | −1.08 ± 0.21 | 97.62 | 665.35 ± 64.89 | 78.71 | 3443.30 ± 240.68 | 100 | 4257.80 ± 135.90 | 88.18 | 91.71 |
| RMSprop | 7686.67 ± 859.43 | 94.96 | −0.67 ± 0.22 | 100 | 798.00 ± 118.66 | 94.40 | 3031.13 ± 410.22 | 88.03 | 4585.67 ± 538.44 | 94.96 | 94.47 |
| Adan | 6480.00 ± 702.14 | 80.05 | −0.91 ± 0.23 | 98.58 | 638.35 ± 18.08 | 75.51 | 2106.90 ± 110.74 | 61.19 | 2665.00 ± 708.69 | 55.19 | 74.11 |
| Lion | 1392.00 ± 139.66 | 17.20 | −1.46 ± 0.65 | 95.42 | 508.15 ± 91.28 | 60.11 | 2432.35 ± 234.61 | 70.64 | 3768.00 ± 520.95 | 78.03 | 64.28 |
| Grams | 7908.00 ± 687.61 | 97.69 | −0.93 ± 0.24 | 98.48 | 633.80 ± 23.52 | 74.97 | 2234.40 ± 130.73 | 64.89 | 3670.12 ± 634.75 | 76.00 | 82.41 |
| Ano (Ours) | 8095.00 ± 494.70 | 100 | −0.97 ± 0.14 | 98.26 | 845.35 ± 56.39 | 100 | 2813.00 ± 243.04 | 81.69 | 4828.80 ± 386.73 | 100 | 95.99 |
| Anolog (Ours) | 7485.00 ± 1010.66 | 92.46 | −0.98 ± 0.14 | 98.19 | 751.05 ± 74.60 | 88.84 | 2983.00 ± 236.42 | 86.63 | 4773.75 ± 602.39 | 98.86 | 93.00 |
| *Tuned* | | | | | | | | | | | |
| Adam | 6430.00 ± 864.51 | 74.55 | −0.98 ± 0.20 | 99.14 | 549.75 ± 51.42 | 66.39 | 406.90 ± 105.28 | 14.40 | 4486.62 ± 683.06 | 75.27 | 65.95 |
| RMSprop | 0.00 ± 452.43 | 0.00 | −0.83 ± 0.14 | 100 | 47.50 ± 96.74 | 5.74 | 16.20 ± 1.47 | 0.57 | 72.50 ± 54.93 | 1.22 | 21.51 |
| Adan | 4840.00 ± 2601.35 | 56.12 | −0.95 ± 0.23 | 99.30 | 754.20 ± 51.44 | 91.08 | 2647.20 ± 534.22 | 93.71 | 4524.75 ± 448.74 | 75.91 | 83.22 |
| Lion | 1324.00 ± 218.94 | 15.35 | −2.38 ± 1.03 | 90.98 | 574.55 ± 73.33 | 69.38 | 2232.85 ± 364.56 | 79.04 | 3759.75 ± 808.26 | 63.07 | 63.57 |
| Grams | 7715.00 ± 627.92 | 89.45 | −1.35 ± 0.75 | 96.95 | 690.40 ± 76.27 | 83.37 | 1989.15 ± 201.64 | 70.42 | 5049.25 ± 624.43 | 84.71 | 84.98 |
| Ano (Ours) | 8625.00 ± 1870.44 | 100 | −0.91 ± 0.21 | 99.55 | 828.10 ± 67.66 | 100 | 2824.85 ± 226.30 | 100 | 5960.88 ± 912.36 | 100 | 99.91 |
| Anolog (Ours) | 1470.00 ± 1176.65 | 17.04 | −1.06 ± 1.06 | 98.67 | 543.60 ± 111.65 | 65.64 | 832.50 ± 125.46 | 29.47 | 1323.50 ± 1694.70 | 22.20 | 46.61 |
| *Best Version* | | | | | | | | | | | |
| Adam | 7615.00 ± 1299.89 | 88.29 | −1.08 ± 0.21 | 97.62 | 665.35 ± 64.89 | 80.35 | 3443.30 ± 240.68 | 100 | 4257.80 ± 135.90 | 71.43 | 87.54 |
| RMSprop | 7686.67 ± 859.43 | 89.12 | −0.67 ± 0.22 | 100 | 798.00 ± 118.66 | 96.37 | 3031.13 ± 410.22 | 88.03 | 4585.67 ± 538.44 | 76.93 | 90.09 |
| Adan | 4840.00 ± 2601.35 | 56.12 | −0.95 ± 0.23 | 98.35 | 754.20 ± 51.44 | 91.08 | 2647.20 ± 534.22 | 76.88 | 4524.75 ± 448.74 | 75.91 | 79.67 |
| Lion | 1392.00 ± 139.66 | 16.14 | −1.46 ± 0.65 | 95.42 | 508.15 ± 91.28 | 61.36 | 2432.35 ± 234.61 | 70.64 | 3768.00 ± 520.95 | 63.21 | 61.36 |
| Grams | 7715.00 ± 627.92 | 89.45 | −1.35 ± 0.75 | 96.03 | 690.40 ± 76.27 | 83.37 | 1989.15 ± 201.64 | 57.77 | 5049.25 ± 624.43 | 84.71 | 82.26 |
| Ano (Ours) | 8625.00 ± 1870.44 | 100 | −0.91 ± 0.21 | 98.60 | 828.10 ± 67.66 | 100 | 2824.85 ± 226.30 | 82.04 | 5960.88 ± 912.36 | 100 | 96.13 |
| Anolog (Ours) | 7485.00 ± 1010.66 | 86.78 | −0.98 ± 0.14 | 98.19 | 751.05 ± 74.60 | 90.70 | 2983.00 ± 236.42 | 86.63 | 4773.75 ± 602.39 | 80.08 | 88.48 |

Table 11: Comparison of the average performance ($\pm$ IC95%) and normalized scores of different optimizers across Atari environments.

## F.2 SAC Settings

| Hyperparameter | Value |
|---|---|
| Total training steps | 1,000,000 |
| Discount $\gamma$ | 0.99 |
| Soft update rate $\tau$ | 0.005 |
| Replay buffer size | $10^6$ |
| Batch size | 256 |
| Learning starts | 5,000 steps |
| Actor LR / Critic LR | see Tab 12 |
| Policy update frequency | 2 |
| Target network update freq. | 1 |
| Entropy coeff. $\alpha$ (init) | 0.2 |
| Entropy autotune | ✓ |
| Max grad. norm (actor) | 0.5 |
| Logging interval | 2048 steps |

Table 13: SAC hyperparameters used in our MuJoCo experiments (values taken from the code).

## F.3 PPO Settings

| Hyperparameter | Value |
|---|---|
| Total timesteps | 10,000,000 |
| Number of envs ($N_{\text{env}}$) | 64 |
| Steps per rollout ($N_{\text{steps}}$) | 64 |
| Batch size ($N_{\text{env}} \times N_{\text{steps}}$) | 4096 |
| Minibatches | 4 |
| Minibatch size | 1024 |
| Update epochs | 4 |
| Discount $\gamma$ | 0.99 |
| GAE $\lambda$ | 0.95 |
| Learning rate | see Tab 12 |
| LR annealing | linear (enabled) |
| Advantage normalization | ✓ |
| Policy clip coef. | 0.10 |
| Value clip | ✓ |
| Entropy coef. | 0.01 |
| Value loss coef. | 0.5 |
| Max grad-norm | 0.5 |
| Target KL | none |

Table 14: PPO hyperparameters used in our Atari experiments (defaults from code).

