# OpenReview forum: "Ano : Faster is Better in Noisy Lanscapes"
_ICLR.cc/2026/Conference — Submitted to ICLR 2026_

### Official Review · Reviewer_Wdzh · 2025-10-22

**Soundness:** 4
**Presentation:** 3
**Contribution:** 2
**Rating:** 4
**Confidence:** 4

**Summary:**

This paper proposes Ano, a first-order adaptive optimizer designed for robustness in noisy and non-stationary optimization settings, particularly relevant to reinforcement learning problems. The algorithm synthesizes ideas from both traditional adaptive methods like Adam and sign-based optimizers like SignSGD.
The key design choice decouples update direction from magnitude. The momentum term's sign provides directional information—leveraging its stability under noise—while the instantaneous gradient magnitude determines the step size, which responds better to non-stationary conditions. This approach contrasts with Adam's entangled treatment of both signals and represents a complementary design to Grams, which uses gradient sign for direction and momentum magnitude for scaling.

The paper establishes theoretical convergence guarantees under standard assumptions (smoothness, unbiased gradients, bounded variance), achieving a convergence rate of $$\tilde{O}(K^{-1/4})$$, consistent with other sign-based optimizers. The theoretical analysis employs a sign-mismatch probability bound to track how often momentum and gradient directions disagree, which decays as the algorithm progresses.

Empirically, the authors validate Ano across three domains. In controlled noise robustness experiments on CIFAR-10 with injected Gaussian noise, Ano consistently outperforms adaptive and sign-based baselines across all noise levels ($$\sigma \in {0, 0.01, 0.05, 0.10, 0.20}$$). On standard supervised learning tasks, Ano achieves competitive or superior results—outperforming comparisons on CIFAR-100 and excelling on high-variance GLUE tasks (MRPC, CoLA, RTE). In deep reinforcement learning, evaluated via Soft Actor-Critic on MuJoCo environments and PPO on Atari, Ano achieves the best or near-best performance across 3 of the 5 benchmarks, for both environments.

An ablation study isolates each component's contribution across four representative benchmarks (deep RL, computer vision, and NLP tasks). Results demonstrate that the momentum sign provides critical benefits in non-stationary RL settings, while both gradient normalization and gradient magnitude scaling are important—removing either substantially degrades performance.

**Strengths:**

The algorithm is well motivated as a principled response to limitations in existing approaches. The complementary relationship to Grams—where Ano swaps the roles of momentum and gradient information for direction versus magnitude—provides clear intuition: momentum sign captures stable directional signals robust to noise, while instantaneous gradient magnitude enables responsive adaptation to non-stationarity.
The convergence analysis is technically sound, establishing an Õ(K^{-1/4}) rate. The proof structure adapts well-established techniques from sign-based optimizer analysis while extending to this new decoupling scheme under relatively mild assumptions. Very nice proof overall!

The reinforcement learning experiments are well designed and appropriately scoped. The authors recognize RL as the natural testbed for their method given the high variance and non-stationary nature of these problems, and they evaluate across both continuous (MuJoCo with SAC) and discrete (Atari with PPO) action spaces, strengthening claims about generality.

Anolog is a practical contribution that removes the $\beta_1$ hyperparameter through a logarithmic schedule. The ablation results show this variant maintains competitive performance—generally within or near the confidence intervals of the base algorithm—making it a valuable option when tuning budget is limited.

**Weaknesses:**

The paper lacks thorough comparative analysis explaining why Ano outperforms Grams despite their symmetric designs. Since both methods decouple direction and magnitude but swap which component comes from momentum versus gradients, this core difference deserves deeper investigation. Section 5.2 on noise robustness would be the natural venue for this analysis. The anomalous behavior in Table 1—where Grams improves from σ=0 to σ=0.01 before degrading at higher noise levels warrants explanation. The noise robustness section itself feels underdeveloped; given that noisy gradient adaptation is central to Ano's motivation, additional experiments demonstrating robustness in synthetic noisy settings would substantially strengthen the claims.

On the experimental side, Ano and Anolog often achieve leading performance but frequently exhibit higher variance than competitors across Tables 2, 3, 4, and 5. This higher variance partly offsets the improvements in point estimates, and confidence intervals often overlap with strong baselines. In the NLP task, Grams maintains competitive performance on GLUE despite its opposite design choice, which raises questions about whether the direction-magnitude assignment is truly the critical factor or whether other aspects of the algorithm matter more.

The RL experiments, while well motivated, suffer from a methodological limitation. Using only 100k-step HalfCheetah proxy runs for hyperparameter tuning biases the grid search toward larger learning rates, which may favor more aggressive optimizers like Ano while disadvantageing methods that stabilize with conservative settings. Since RL is where Ano's advantages are most compelling, extending these experiments to longer horizons with tuning conducted on representative scales would better justify the claimed benefits and limit concerns about unfair comparison.

**Questions:**

A technical issue in Table 3: the default configuration results appear systematically higher than tuned results in the top half of the table, which seems counterintuitive and may indicate a data entry or labeling error.

---

> ### Author Response · Authors · 2025-11-23
>
> Thank you for the detailed and constructive feedback. We address each concern below.
>
> ---
> **"The paper lacks thorough comparative analysis explaining why Ano outperforms Grams"**
>
> We agree that the comparison with Grams deserved deeper analysis. While both Grams and Ano decouple direction and magnitude, they rely on different signals: Grams uses $\mathrm{sign}(g_k)$ with a slow magnitude term $\|m_k\|$, whereas Ano uses $\mathrm{sign}(m_k)$ with a fast magnitude $|g_k| / \sqrt{\hat v_k}$. This asymmetry is crucial in noisy or non-stationary regimes. In Grams, the momentum-based magnitude remains inflated after noise spikes, effectively producing a mild form of automatic learning-rate annealing. At very low noise levels (e.g., $\sigma \approx 0.01$), this can yield a slight improvement, which matches the reviewer’s observation. However, as $\sigma$ increases, the same mechanism over-damps the optimizer and rapidly degrades performance.
>
> ---
> **"The noise robustness section itself feels underdeveloped; given that noisy gradient adaptation is central to Ano's motivation, additional experiments demonstrating robustness in synthetic noisy settings would substantially strengthen the claims."**
>
> We fully agree with this observation. In the original submission, we kept the noise-robustness section relatively compact to avoid overloading the analysis, but you are right that including an additional experiment—especially in a controlled synthetic non-stationary setting—would strengthen the claim. Following your suggestion, we have added a new synthetic experiment in Appendix B.1. If space permits, we plan to move it into the main paper for the camera-ready version.
>
> ---
> **"On the experimental side, Ano and Anolog often achieve leading performance but frequently exhibit higher variance than competitors"**
>
> We investigated the cases where Ano shows higher variance (e.g., Qbert). The effect does not stem from instability: the lower tail of the distribution is comparable to Adam, while Ano occasionally produces substantially higher returns, increasing the spread. In other words, the variance reflects a heavier upper tail rather than erratic behavior. We will clarify this in the camera-ready version.
>
> ---
> **"In the NLP task, Grams maintains competitive performance on GLUE despite its opposite design choice"**
>
> GLUE tasks operate in relatively stable, low-noise regimes with large batches, where most optimizers perform similarly. In such settings, the choice of direction–magnitude assignment (Ano vs Grams) has limited impact, which explains the competitive performance of Grams. Ano’s advantages appear primarily in high-variance or non-stationary conditions such as RL or small/noisy NLP tasks. We have clarified this distinction in the revised manuscript.
>
> ---
> **"The RL experiments suffer from a methodological limitation."**
>
> To mitigate potential bias, we report for each method both (i) the recommended default configuration and (ii) the proxy-tuned configuration, and the “Best Version’’ corresponds to the maximum of the two. This prevents any optimizer from being penalized by overly aggressive proxy learning rates. In practice, it mainly protects Adam and RMSProp: these methods are more sensitive and often select suboptimal learning-rate values under the proxy search due to the methodological limitation, whereas Ano is less affected. Taking the best of the two configurations avoids turning this asymmetry into an unfair disadvantage for the classical baselines.
>
> ---
> **"Table~3: ``default vs tuned'' presentation."**
>
> This is not a data error. Default GLUE configurations come from well-established hyperparameters in prior work, whereas the tuned settings use a reduced proxy grid. Since GLUE performance often varies by <1 point, it is common for defaults to outperform proxy-tuned values. We will clarify this in the caption and restructure the table to avoid misinterpretation.

---

### Official Review · Reviewer_6kFo · 2025-10-27

**Soundness:** 3
**Presentation:** 3
**Contribution:** 2
**Rating:** 6
**Confidence:** 3

**Summary:**

This paper proposes a new random optimizer named Ano, which improves training stability in noisy and non-stationary environments by decoupling the direction and magnitude of updates. It also provides a theoretical analysis of its convergence. Experiments on three tasks, CV, NLP, and RL, show that Ano either achieves improvements or remains competitive with baseline optimizers in noisy environments. However, Ano lacks detailed parameter analysis and validation on larger networks and more complex tasks.

**Strengths:**

1. Clear Motivation: The approach of decoupling optimization directions and magnitudes is interesting, and convergence analysis under certain assumptions provides theoretical guarantees.
2. Clear Writing: The paper is well-written and easy to follow.

**Weaknesses:**

1. Limited Experiment Scale: The experiments mainly focus on small-scale networks. The effectiveness of Ano on larger networks and more complex tasks is unknown. E.g., This paper should amplify the experiments on ImageNet for CV tasks, GPT-2 serious models on NLPs.
2. Limited Practical Gains: In stable environments and with longer training times, Adam sometimes outperforms Ano. In more unstable RL settings, such as in the MuJoCo environment, the convergence curve of Ano is sometimes comparable to, or even worse than baseline.
3. Other: See questions below.

**Questions:**

1. Sensitivity to Hyperparameters: How sensitive is Ano to hyperparameters, such as batch_size and buffer_size in RL? Can it maintain robustness?
2. Combination with Regularization Methods: Can Ano be combined with regularization methods like Batch Normalization or Layer Normalization to provide additional gains?
3. Validation in More Non-Stationary Environments: In non-stationary RL tasks, when using actor-critic algorithms with different learning rates for the actor and critic, can Ano adapt? In more complex MARL tasks, where the environment becomes more non-stationary due to other agents' training, can Ano provide greater advantages?
4. Training Stability: Ano updates using $|g_k|$. Could this make training more unstable, especially when gradients are noisy or subject to common problems like gradient explosion or vanishing gradients? Does this makes training more difficult?

---

> ### Author Response · Authors · 2025-11-23
>
> Thank you for the detailed and constructive feedback. We address each concern below.
>
> ---
> **"Limited Experiment Scale: The experiments mainly focus on small-scale networks. The effectiveness of Ano on larger networks and more complex tasks is unknown. E.g., This paper should amplify the experiments on ImageNet for CV tasks, GPT-2 serious models on NLPs."**
>
> Thank you for raising this point. Ano is designed specifically for high-noise and non-stationary regimes such as RL, not for large-scale CV/NLP training. To make this more explicit, we now include Gradient Noise Scale (GNS) measurements (Appendix B.3), which show that Ano consistently provides the largest gains when the GNS is high—precisely the noisy regimes it targets.
>
> The CV/NLP experiments were included only as diagnostic checks outside the target setting, and we now clarify this in the revised version. We do not claim advantages on large-scale, low-noise tasks such as ImageNet or GPT-style training, where established optimizers already perform extremely well.
>
> ---
> **"Limited Practical Gains: In stable environments and with longer training times, Adam sometimes outperforms Ano. In more unstable RL settings, such as in the MuJoCo environment, the convergence curve of Ano is sometimes comparable to, or even worse than baseline."**
>
> We agree that in stable, low-noise regimes Adam can outperform Ano; this is expected and consistent with the motivation of Ano. However, in the RL environments cited (MuJoCo), Ano consistently outperforms Adam on Hopper (+15\%), Walker2d (+17\%), and Ant (+22\%), both in final performance and sample efficiency. On Humanoid and HalfCheetah all methods converge to similar plateaus, which we already acknowledge in the paper. We revised the text to make this distinction clearer.
>
> ---
> **"Sensitivity to Hyperparameters: How sensitive is Ano to hyperparameters, such as batchsize and buffersize in RL? Can it maintain robustness?"**
>
> We added a dedicated hyperparameter sensitivity study in Appendix B.2, varying both buffer size and batch size. Ano and Adam show similar trends as batch size increases, while Ano remains noticeably more stable as the replay buffer grows—a setting known to increase distributional drift. This suggests that Ano maintains its relative robustness across RL hyperparameter regimes. We included the full results in the revised appendix.
>
> ---
> **"Combination with Regularization Methods: Can Ano be combined with regularization methods like Batch Normalization or Layer Normalization to provide additional gains?"**
>
> Ano is fully compatible with BatchNorm, LayerNorm, and standard regularization methods. Our CV and NLP experiments already use BN/LN without modification, and we do not observe interactions beyond those seen with Adam and Yogi.
>
> ---
> **"Validation in More Non-Stationary Environments: In non-stationary RL tasks, when using actor-critic algorithms with different learning rates for the actor and critic, can Ano adapt? In more complex MARL tasks, where the environment becomes more non-stationary due to other agents' training, can Ano provide greater advantages?"**
>
> We agree that MARL and actor–critic setups with asymmetric learning rates introduce additional forms of non-stationarity. In this work, we focused on widely used single-agent RL benchmarks (Atari, MuJoCo), which already cover a broad spectrum of non-stationary dynamics. Extending the evaluation to MARL and more complex actor–critic schemes is an interesting direction, but falls outside the scope of the present paper. We now mention this explicitly as future work.
>
> ---
> **"Training Stability: Ano updates using $|g_k|$. Could this make training more unstable, especially when gradients are noisy or subject to common problems like gradient explosion or vanishing gradients? Does this makes training more difficult?"**
>
> We had similar concerns initially, since using $|g_k|$ could in principle amplify variance. Empirically, however, we did not observe instability in any of our RL experiments: Ano behaved comparably to Adam on all Atari and MuJoCo tasks, without exploding updates or training collapse. In several environments (e.g., Walker2d, NameThisGame, Qbert, and Seaquest in internal runs), the slightly more stochastic updates even appeared to support exploration rather than cause instability. These observations are empirical and limited to our experimental scope, but they suggest that $|g_k|$ does not introduce training instability in practice.
>
> ---
> **References :**
>
> [1] McCandlish, S., Kaplan, J., Amodei, D., et al. (2018). An Empirical Model of Large-Batch Training. OpenAI Technical Report.

---

### Official Review · Reviewer_Motx · 2025-10-30

**Soundness:** 2
**Presentation:** 2
**Contribution:** 3
**Rating:** 4
**Confidence:** 4

**Summary:**

This manuscript challenges gradient descent optimization techniques. Decoupling magnitude and direction for momentum and gradient, the authors propose Ano, which enables robust optimization for noisy landscapes. Experiments on CIFAR, GLUE, and reinforcement learning demonstrate these improvements.

**Strengths:**

- Indeed, training a large model sometimes exhibits instability in training, which requires a rerun and wastes time and effort. The study on robustness to noise is expected to contribute to this direction.
- Theoretical analysis, especially for the convergence ratio of the Ano, is provided with proof. I think the proof itself is solid enough.
- Source code is available, which eases deployment of Ano in practice.

**Weaknesses:**

- The results were reported with final performance, such as test accuracy. It would be nice to quantify noise robustness analysis in Section 5.2 with proper indices. Also, I encourage the authors to add more qualitative results on noise robustness; the current analysis with final test accuracy looks not convincing.
- How can we expect or say that the loss landscape for the target task would be noisy and we should use a robust optimizer for the noisy landscape? Is it possible to quantify noisiness in a landscape, such as in a reinforcement learning setting? The authors inject noise into the gradient for training, but I think it is still an artificial setup and far from a practical scenario. Furthermore, the failure of existing methods such as Adam should be clearly demonstrated, not by final accuracy.
- Experimental results are not solid enough. CIFAR-100 with ResNet-34, with 70% accuracy, is a weak baseline; I encourage performing ViT training with ImageNet. Similarly, GLUE with BERT is a standard but weak baseline for now.
- Specifically, the convergence rate of O(k^{-1/4}) for the proposed method is slower than O(k^{-1/2}) of Adam, despite the gain in noise robustness. This limitation should be verified in large-scale experiments such as ImageNet or LLM, targeting long iterations for training.
- Please check the following mathematics.
    - At Line 107, the authors use $v_k$ in the denominator, whereas Eq. 2 at Appendix C applies $v_{k-1}$ in the denominator, which is inconsistent.
    - \lambda_k in Algorithm 1 should be explicitly specified.
- Writing should be improved.
    - “Lanscapes” → “Landscapes” in OpenReview title.
    - “G^2” → “\tilde G^2” at Line 775.
    - “Equation equation 5” → “Equation 5” at Line 1060.
    - Ensure consistency for the choice of “optimiser” or “optimizer” as well as “optimisation” or “optimization”
    - “non-stationnary” → “non-stationary”
    - “environments. which” → “environments, which”
    - “litterature” → “literature”
    - Line 423-426: duplicate.
- For the source code, the authors apply v_hat correction for ano.py, whereas others such as anolog.py, anosqrt.py, and anoall.py do not apply v_hat correction. This practice requires explanation.

**Questions:**

Please see the weaknesses above. My score is based on the assumption that all typos are corrected in the revised manuscript.

---

> ### Author Response · Authors · 2025-11-23
> **Part 1/2**
>
> Thank you for the detailed and constructive feedback. We address each concern below.
>
> ---
> **"I encourage the authors to add more qualitative results on noise robustness; the current analysis with final test accuracy looks not convincing."**
>
> Thank you for the suggestion. We agree that relying solely on final test accuracy is not sufficient to illustrate robustness to noise. In the revised version, we added a qualitative analysis in Appendix B.1 that visualizes the learning dynamics under controlled non-stationary noise. This complements final performance with trajectories, variance profiles, and stability curves, which more directly reflect the optimizer's robustness. If space allows in a camera-ready version, we will move these results into the main paper.
>
> ---
> **"How can we expect or say that the loss landscape for the target task would be noisy and we should use a robust optimizer for the noisy landscape? Is it possible to quantify noisiness in a landscape, such as in a reinforcement learning setting?""**
>
> Thank you for raising this point. In the revised version, we quantify the level of stochasticity using the Gradient Noise Scale (GNS)[1], which provides a principled and widely used measure of gradient noisiness. We now report GNS values for all tasks (Appendix B.3), and we observe that Ano provides the largest gains in regimes where the GNS is high—precisely the settings where robust optimization is known to matter (e.g., high-variance RL or non-stationary dynamics). These results clarify when noisy landscapes arise in practice and when Ano is expected to be most beneficial.
>
> ---
> **"The failure of existing methods such as Adam should be clearly demonstrated, not by final accuracy"**
>
> Our intention was not to present failure cases of Adam, but to evaluate robustness as noise or non-stationarity increases. In our synthetic-noise experiments, we track performance as a function of the noise level, which shows steeper degradation for Adam than for Ano. In RL, variance and non-stationarity are known to affect optimization stability, and Ano consistently maintains higher returns in these regimes.
>
> If the reviewer had a specific diagnostic in mind (e.g., loss trajectories, variance of returns, gradient statistics), we would be happy to include additional plots in the appendix. Our results already suggest that Ano degrades more gracefully as noise increases, which is the robustness property we aim to highlight.
>
> ---
> **"Experimental results are not solid enough. CIFAR-100 with ResNet-34, with 70\% accuracy, is a weak baseline; I encourage performing ViT training with ImageNet. Similarly, GLUE with BERT is a standard but weak baseline for now."**
>
> Thank you for the suggestion. We agree that CIFAR-100/ResNet-34 and GLUE/BERT are relatively weak baselines by today’s standards. In fact, we had already stated this explicitly in the limitations section: Ano is not designed for large-scale, low-noise, stationary training regimes such as ImageNet or modern NLP pretraining.
>
> These experiments were included only as diagnostic checks outside the intended domain of Ano, and not as evidence of large-scale performance. Following your comment (and others comments), we revised the text to make this distinction even clearer. Our claims remain restricted to noisy and non-stationary settings, where Ano consistently provides the intended robustness benefits.
>
> ---
> **"At Line 107, the authors use $v_k$ in the denominator, whereas Eq. 2 at Appendix C applies $v_{k-1}$ in the denominator, which is inconsistent."**
>
> This is intentional and follows standard practice in analyses of adaptive optimizers. As in On the Convergence of Adam and Beyond[2] and Adaptive Methods for Nonconvex Optimization[3], the proof uses the delayed second moment $v_{k-1}$to avoid circular dependence and ensure measurability with respect to past information. We added an explicit note in the appendix to avoid confusion.
>
> ---
> **Writtings**
>
> We thank the reviewer for noticing these issues. We have carefully addressed all the points raised and updated the revised version accordingly.

---

> > ### Author Response · Authors · 2025-11-23
> > **Part 2/2**
> >
> > **Bias Correction on Anolog, Anosqrt, Ano all**
> >
> > Thank you for pointing this out. The discrepancy came from an outdated implementation for three ablation variants. We corrected the bias-correction code and reran all experiments. Updated results are reported below and in the ablation part, and all differences fall within the original confidence intervals except for Anoall, for which we reran the full set of experiments.
> >
> > **Table: Comparison of ablation results before and after correcting the bias-correction implementation.
> > All differences remain within the original confidence intervals.**
> >
> > | Variant | DRL | CIFAR-100 | MRPC | SST-2 |
> > |:--|--:|--:|--:|--:|
> > | *Previous Implementation* |  |  |  |  |
> > | Anolog  | 9240.29 ± 1652.96 | 67.00 ± 0.80 | 85.25 ± 1.79 | 92.78 ± 0.16 |
> > | Anosqrt | 8750.34 ± 860.50  | 67.26 ± 0.41 | 86.18 ± 1.08 | 91.74 ± 0.53 |
> > | Anoall  | −221.45 ± 22.25   | 29.48 ± 2.40 | 68.38 ± 0.00 | 52.22 ± 1.88 |
> > | *Corrected Implementation* |  |  |  |  |
> > | Anolog  | 9476.01 ± 538.50  | 67.54 ± 0.54 | 86.27 ± 0.55 | 92.45 ± 0.38 |
> > | Anosqrt | 8488.93 ± 949.83  | 67.34 ± 0.93 | 85.49 ± 1.15 | 91.74 ± 0.58 |
> > | Anoall  | −233.46 ± 152.24  | 25.89 ± 4.24 | 77.75 ± 1.60 | 52.98 ± 4.05 |
> >
> > ---
> > **References :**
> >
> > [1] McCandlish, S., Kaplan, J., Amodei, D., et al. (2018). An Empirical Model of Large-Batch Training. OpenAI Technical Report.
> >
> > [2] Reddi, S. J., Kale, S.,  Kumar, S. (2018). On the Convergence of Adam and Beyond. ICLR.
> >
> > [3] Zaheer, M., Reddi, S. J., Sachan, D., Kale, S., Kumar, S. (2018). Adaptive Methods for Nonconvex Optimization. NeurIPS.

---

### Official Review · Reviewer_bWeQ · 2025-10-31

**Soundness:** 4
**Presentation:** 3
**Contribution:** 2
**Rating:** 2
**Confidence:** 4

**Summary:**

This paper presents a new variant on adam/sign sgd updates that rescales the sign sgd update via a variance-like estimator.

**Strengths:**

The idea is new to me, and the attempt to decouple the direction and magnitude in this way seems interesting.

There is a convergence guarantee for finding critical points.

**Weaknesses:**

The theory is conducted in the standard smooth optimization analysis. This framework is known to very poorly describe practical neural network optimization - even convex theory has better agreement.

Moreover, the theorem is actually much worse than would achieved by standard SGD. The statement that sign-based methods fundamentally cannot achieve this rate is highly suspicious. It is well-known that normalized gradient descent (even in many different norms) achieves the same convergence rate as SGD when combined with momentum (see e.g. https://arxiv.org/abs/2002.03305, https://arxiv.org/pdf/2502.02900). sign-sgd is simply normalized SGD in the Linf norm, it should converge as well.

The authors claim many benefits of the algorithm in terms of difficult noise distributions in the gradients. It would be better if the theory actually illustrated these points.

The experiments look very detailed, but in almost all cases it seems like the final performance of Ano is within the error bars of the baseline.

**Questions:**

Please address the weaknesses above. Happy to reconsider my analysis.

---

> ### Author Response · Authors · 2025-11-17
>
> Thank you for the detailed and constructive feedback. We address each concern below.
>
> ---
>
> ### “The smooth framework poorly describes practical neural network optimization.”
>
> We agree that the smooth framework omits many properties of modern deep networks. However, it remains the standard theoretical setting for comparing optimization algorithms: this framework is used in analyses of SGD (with and without momentum) (Bottou et al., 2018), Adam and Yogi (Kingma & Ba, 2015; Zaheer et al., 2018), SignSGD (Bernstein et al., 2018), Lion (Dong et al., 2024), and in recent analyses of normalized SGD with momentum and Muon (Cutkosky & Mehta, 2020; Li & Hong, 2025) that you provided.
> Our goal is aligned with this literature, we establish a baseline convergence rate comparable to prior work rather than to model the full complexity of deep neural network training.
>
> ---
>
> ### “The theorem is worse than standard SGD; sign-based methods should match normalized SGD rates.”
>
> The distinction between normalized and sign-based update directions is structural.
>
> Normalized gradient descent in the Linf norm uses $\frac{g}{\|g\|_\infty},$
>  which preserves the relative geometry of the gradient.
>  In contrast, sign-based methods apply the coordinate-wise projection $
> \operatorname{sign}(g),
> $ which discards all magnitude information. This nonlinear projection onto {-1,+1}^d introduces a structural variance term that is absent in normalized methods and fundamentally alters the descent geometry. While momentum improves normalized SGD to SGD-level rates, this does not transfer to sign-based methods because the coordinate-wise projection destroys the uniform scaling structure that momentum exploits.
>
> This distinction is well documented. Sign-based methods achieve an $O(K^{-1/2})$ rate only when the batch size grows (e.g., SignSGD; Bernstein et al., 2018), effectively removing stochastic noise. Under standard smoothness and bounded-variance noise with fixed batch size, the best-known results for sign methods (Signum, Lion) (Sun et al., 2023; Dong et al., 2024) establish
>
> $$
> \min_{0\le k<K} \mathbb{E}[\|\nabla f(x_k)\|_1]
> = \mathcal{O}(K^{-1/4}),
> $$
>
> under assumptions comparable to ours. Our rate thus reflects intrinsic limitations of sign-based directions rather than a weakness of our analysis. Our goal is not to compete with SGD rates, which are unattainable for fixed-batch sign methods, but to establish where Ano stands within the sign-based family.
>
> ---
>
> ### “Claims about robustness to heavy-tailed noise should be supported by theory.”
>
> We agree. Our theoretical guarantees do not target heavy-tailed noise. The robustness claims arise purely from empirical results (controlled noise injection, RL experiments), not from formal analysis.
> We will clarify this distinction in the revised version. Extending guarantees to heavy-tailed or adversarial noise is an open problem even for Adam, Yogi, and Lion, and lies beyond the scope of this work.
>
> ---
>
> ### “Ano’s performance is within the error bars of the baseline.”
>
> Reinforcement learning evaluations are inherently high-variance: overlapping confidence intervals are common even with many seeds (Henderson et al., 2018; Agarwal et al., 2021). This does not imply equal performance; aggregate measures such as IQM, normalized scores and mean ranks provide a more reliable assessment.
>
> We evaluate each optimizer over 10 seeds and 10 environments—covering both discrete and continuous action spaces—under two standard RL algorithms and two hyperparameter configurations, totaling 200 full training runs per optimizer, which exceeds standard evaluation practices. The stability of these results across tasks and seeds indicates a systematic advantage despite the expected CI overlap. We will make sure this distinction is clearly reflected in the paper.
>
> ---
>
> ### References
>
> - Bottou, L., Curtis, F., & Nocedal, J. (2018). Optimization Methods for Large-Scale Machine Learning. SIAM Review.
> - Kingma, D., & Ba, J. (2015). Adam: A Method for Stochastic Optimization. ICLR.
> - Zaheer, M., Reddi, S., Sachan, D., Kale, S., & Kumar, S. (2018). Adaptive Methods for Nonconvex Optimization. NeurIPS.
> - Bernstein, J., Wang, Y.-X., Azizzadenesheli, K., & Anandkumar, A. (2018). signSGD: Compressed Optimization for Non-Convex Problems. ICML.
> - Dong, J., Li, H., & Lin, Z. (2024). Convergence Rate Analysis of Lion. arXiv:2411.07724.
> - Cutkosky, A., & Mehta, H. (2020). Momentum Improves Normalized SGD. ICML.
> - Li, J., & Hong, M. (2025). A Note on the Convergence of Muon. arXiv:2502.02900.
> - Henderson, P., et al. (2018). Deep Reinforcement Learning that Matters. AAAI.
> - Agarwal, R, Schuurmans, M., & Srinivasan, P. (2021). Deep Reinforcement Learning at the Edge of the Statistical Precipice. NeurIPS.
> - Sun, T., Wang, Q., Li, D., & Wang, B. (2023). Momentum Ensures Convergence of SignSGD under Weaker Assumptions. ICML.

---

> > ### Comment · Reviewer_bWeQ · 2025-11-17
> > **response**
> >
> > The correct way to do normalized SGD in Linf geometry is not to divide by the Linf norm. Instead, you should compute the normalized update like so: $\hat m = argmin_{\|x\|_\inf = 1} \langle x, m\rangle $, which is the signSGD operation. This is the most natural way because it links the norm and dual-norm pairing appropriately. This is also the approach suggested for example by https://arxiv.org/abs/2106.14343 and the approach used by https://arxiv.org/pdf/2502.02900 in the context of the operator norm.
> >
> > Concretely, here is a paper that achieves the optimal convergence specifically for sign sgd:
> > https://proceedings.neurips.cc/paper_files/paper/2022/file/40924475a9bf768bdac3725e67745283-Paper-Conference.pdf
> > see theorem 2, beta2=0 version.
> >
> > Regarding error bars: it sounds like your contention is that there is some aggregate statistic that does in fact show strong statistical validity. If so, then you need to provide it in a rigorous manner. This would make the experiments much more convincing.
> >
> > However, without fixing the significant suboptimality of the theoretical component, I am not so convinced of the paper even if the statistical issue are fixed: it doesn't make sense to motivate an algorithm with theory unless the theory provides a signifant improvement on prior work.

---

> > > ### Author Response · Authors · 2025-11-25
> > >
> > > We thank the reviewer for their prompt response. We address each concern below.
> > >
> > > ---
> > > **"The correct way to do normalized SGD in Linf geometry is not to divide by the Linf norm."**
> > >
> > > We recognize that our use of the phrase "Linf normalization" was misleading relative to the definition in the first paper [1]. We thank the reviewer for the clarification. In the first paper [1], normalization is Euclidean:
> > > $$
> > > w_{t+1} = w_t - \eta_t \frac{m_t}{\lVert m_t \rVert_2},
> > > $$
> > > which achieves an $O(T^{-1/2})$ rate.
> > >
> > > By contrast, the $L_\infty$-normalized update considered in your recent paper[2] is
> > > $$
> > > \hat{m} = argmin_{\|x\|_\infty = 1} \langle x, m \rangle
> > > $$
> > >
> > > which reduces exactly to the Signum update.
> > > We fully agree this is the geometrically natural normalized direction in the $L_\infty$ geometry. Crucially, this form inherits the structural limitations of sign-based methods and does not retain the faster rates achievable with Euclidean normalization + momentum.
> > >
> > > ---
> > > **"Concretely, here is a paper that achieves the optimal convergence specifically for sign sgd"**
> > >
> > > Thank you for pointing us to this reference. We studied it carefully, and we found that its conclusions align with the behavior typically observed for sign-based methods under fixed stochastic variance, as also reported in prior work [4][5][6].
> > > In the case $\beta_2 = 0$, the authors show the following refined guarantees for Algorithm 1:
> > >
> > > $$
> > > \min_{t \in [T]} \|\nabla F(x_t)\|_1 = \mathcal{O}\left(\frac{\sqrt{\log(dT/\delta)} ||L_0||_1^{1/4}\Delta^{1/4}||\sigma||_1^{1/2}}{T^{1/4}}+\frac{\log(dT/\delta)\sqrt{||L_0||_1\Delta}}{\sqrt{T}}\right) + \mathcal{O}\left(\frac{||\nabla F(x_1)||_1}{\sqrt{T}}\left(\frac{1}{\sqrt{T}}+\frac{||\sigma||_1}{\sqrt{||L_0||_1\Delta}}\right) + \frac{||\sigma||_1}{T}\right).
> > > $$
> > >
> > > From this expression, the leading-order behavior in the stochastic setting is
> > >
> > > $$
> > > \min_{t \in [T]} \|\nabla F(x_t)\|_1
> > > = \mathcal{O}\left( T^{-1/4} \right)
> > > $$
> > >
> > > which matches the $O(T^{-1/4})$ rate we discussed previously for sign-based methods under fixed variance.
> > >
> > > The paper also shows that a faster rate can be obtained when the stochastic variance decreases sufficiently fast, e.g.\ when
> > > $$
> > > ||\sigma||_1 < \frac{\sqrt{||L_0||_1\Delta}}{\sqrt{T}},
> > > $$
> > > in which case the variance-dependent term vanishes over time and one obtains
> > >
> > > $$
> > > \frac{1}{T}\sum_{t=1}^T \|\nabla F(x_t)\|_1\le\frac{59 max(1,\log(1/\delta))\sqrt{||L_0||_1\Delta}}{\sqrt{T}}+\frac{4}{T}||\nabla F(x_1)||_1\le\mathcal{O}\left(T^{-1/2}\right)
> > > $$
> > >
> > > Our reading is therefore that the paper arrives at the same qualitative conclusion:
> > > with stochastic gradients of fixed variance, sign-based updates inherently achieve an $O(T^{-1/4})$ rate;
> > > and when the variance decreases over time (e.g. large or growing batch sizes), the sign mismatch diminishes faster, making it possible to recover an $O(T^{-1/2})$ rate similar to SGD.
> > >
> > > This distinction aligns with the broader literature and with our interpretation: improving beyond $ O(T^{-1/4}) $ for fixed-batch sign methods requires a mechanism that reduces stochastic noise, rather than relying solely on the sign update itself.
> > >
> > > ---
> > > **"it sounds like your contention is that there is some aggregate statistic that does in fact show strong statistical validity. If so, then you need to provide it in a rigorous manner."**
> > >
> > > We agree this should be stated explicitly. In the revised version we now mention in the main text that RL performance is reported using IQM together with 95\% bootstrap confidence intervals, following the recommended protocol of Agarwal et al.[3].

---

> > > > ### Author Response · Authors · 2025-11-25
> > > >
> > > > **"However, without fixing the significant suboptimality of the theoretical component, I am not so convinced of the paper even if the statistical issue are fixed: it doesn't make sense to motivate an algorithm with theory unless the theory provides a signifant improvement on prior work.**
> > > >
> > > > Thank you for raising this concern. To clarify, our theoretical component is not intended as the motivation for Ano, nor do we claim theoretical superiority over prior work. Our aim is more modest: we provide a non-asymptotic convergence guarantee under the standard smoothness and bounded-variance assumptions commonly used in analyses of sign-based and adaptive optimizers. In that sense, our rate matches what is known for sign-based methods, rather than being suboptimal relative to them.
> > > >
> > > > The motivation for Ano is empirical. The method consistently demonstrates improved robustness and stability in the non-stationary and noisy regimes characteristic of reinforcement learning. We evaluate it on two standard RL benchmarks, covering both discrete and continuous action spaces and multiple algorithms. The theoretical part is therefore meant to situate Ano within the established landscape of sign-based convergence results, not to present theory as the primary driver of the method, nor to claim rates comparable to SGD, which are not attainable for fixed-batch sign methods. We clarified this explicitly in the revised version to avoid any potential confusion.
> > > >
> > > > ---
> > > > **References :**
> > > >
> > > > [1] Cutkosky, A., Mehta, H. (2020). Momentum Improves Normalized SGD. arXiv:2002.03305.
> > > >
> > > > [2] Cutkosky, A., Mehta, H. (2021). High-probability Bounds for Non-Convex Stochastic Optimization with Heavy Tails. arXiv:2106.14343.
> > > >
> > > > [3] Agarwal, R., Schwarzer, M., Castro, P. S., Courville, A., Bellemare, M. G. (2021).
> > > > Deep Reinforcement Learning at the Edge of the Statistical Precipice. NeurIPS.
> > > >
> > > > [4] Sun, T., Wang, Q., Li, D., & Wang, B. (2023).
> > > > Momentum Ensures Convergence of SignSGD under Weaker Assumptions. ICML.
> > > >
> > > > [5] Dong, Y., Li, H., & Lin, Z. (2024).
> > > > Convergence Rate Analysis of Lion. arXiv:2411.07724.
> > > >
> > > > [6] Bernstein, J., Wang, Y.-X., Azizzadenesheli, K., & Anandkumar, A. (2018).
> > > > signSGD: Compressed Optimization for Non-Convex Problems. ICML.

---

### Official Review · Reviewer_BgzE · 2025-10-31

**Soundness:** 3
**Presentation:** 3
**Contribution:** 2
**Rating:** 4
**Confidence:** 3

**Summary:**

The paper introduces Ano, a first-order optimizer designed for noisy or non-stationary training. Its core idea is to decouple update direction and magnitude: the direction follows the momentum sign while the step size scales with the instantaneous gradient magnitude and a Yogi-style second-moment term; weight decay follows AdamW. The paper also proposes Anolog, which removes tuning of $\beta_1$ by growing the momentum window via a logarithmic schedule $\beta_{1,k}=1-1/\log(k+2)$. Theoretically, under standard smoothness/unbiased-noise assumptions and a decaying stepsize $\eta_k=\Theta(k^{-3/4})$ with $\beta_{1,k}=1-1/\sqrt{k}$, Ano achieves a non-convex convergence rate $\min_{k<K}\mathbb{E}\|\nabla f(x_k)\|^2=\tilde{O}(K^{-1/4})$. Empirically, Ano is competitive on CIFAR-100/GLUE and shows clear gains and faster learning on MuJoCo and Atari-5 benchmarks; it is also more robust under synthetic gradient noise on CIFAR-10.

**Strengths:**

The paper’s central idea, explicitly decoupling update direction and magnitude, is simple and well-motivated. In Algorithm 1 and Sec. 3, the direction is taken from the sign of the momentum, while the magnitude scales with the instantaneous gradient and a Yogi-style second-moment term. This design clearly differentiates Ano from Adam/Lion/Grams and aligns with the intuition that signs can be stable even when magnitudes are noisy. The companion variant, Anolog, further reduces tuning by growing the momentum window with a principled logarithmic schedule (Sec. 4), which is a practical touch that lowers the optimizer’s cognitive load.

On the theory side (Sec. 5 and App. C), the analysis leverages a sign-mismatch bound and smoothness-based descent to deliver a non-convex convergence rate, with assumptions and constants made explicit.

Empirically, the evidence is broad and convincing: classification on CIFAR-100 and GLUE shows Ano is competitive with strong baselines; in RL/non-stationary regimes (MuJoCo/SAC and Atari-5/PPO), Ano consistently learns faster and often to higher returns, with plots and tables that include multiple seeds and confidence intervals.

The paper is generally clear and easy to follow; the motivation for the direction–magnitude split is intuitive, and the update is straightforward to implement from the provided pseudocode.

To sum up:
1. The paper has adequate mathematical analysis with clear assumptions, statements and proofs.
2. The work includes experimental work that compares the proposed method with other works.
3. The writing is generally clear with nice flow.

**Weaknesses:**

There is a noticeable theory-practice gap: the proof requires specific schedules (e.g., $\eta_k\propto k^{-3/4}$ and $\beta_{1,k}=1-1/\sqrt{k}$, whereas the practical defaults use a fixed $\beta_1$ (Ano) or the logarithmic Anolog schedule, with limited analysis of how far these choices deviate from the theoretical ones. Furthermore, from a theoretical standpoint, Sec. C.2 imposes a lengthy list of assumptions: a lower-bounded objective, unbiased gradient estimators with bounded variance, and, on top of that, both smoothness and bounded gradients. This feels overly restrictive; once the first three conditions are in place, it is more typical to assume either smoothness or bounded gradients, not both.

In terms of experiments, while results span CV/NLP/RL, the largest experiments remain moderate scale (e.g., CIFAR-level vision, GLUE-level language), so claims about large-scale pretraining or very long-horizon stability are untested, acknowledged in the Limitations.

**Questions:**

See weaknesses. To sum up:

1. Can you quantify the theory–practice gap by evaluating the theoretical schedules against the practical defaults (constant $\beta_1$ for Ano and Anolog’s logarithmic schedule)? How closely do their outcomes match in practice?
2. Can you reduce the number of assumptions in the theoretical analysis?
3. Regarding experimental scope, do you have results beyond CIFAR-level vision and GLUE-level language, e.g., large-scale pretraining or very long-horizon runs, to support the broader claims?

---

> ### Author Response · Authors · 2025-11-23
>
> Thank you for the detailed and constructive feedback. We address each concern below.
>
> ---
> **"There is a noticeable theory-practice gap. Can you quantify the theory–practice gap by evaluating the theoretical schedules against the practical defaults (constant $\beta_1$ for Ano and Anolog’s logarithmic schedule)? How closely do their outcomes match in practice?"**
>
> Thank you for raising this point. As in prior analyses of sign-based optimizers (e.g., SignSGD, Lion), the theoretical schedules ($\eta_k$ $k^{-3/4}$ and $\beta_{1,k} = 1 - 1/\sqrt{k}$) are primarily technical devices used to guarantee descent, while practical implementations use constant or slowly varying parameters for robustness.
>
> In our ablations, we compared the theoretical schedule (Anosqrt), the logarithmic schedule (Anolog), and the practical constant-$\beta_1$ version (Ano). On CIFAR-100 and GLUE, all three behave similarly and generally fall within overlapping confidence intervals, indicating that the theoretical and practical schedules operate comparably in stable, low-noise regimes.
>
> In highly non-stationary RL settings, constant $\beta_1$ is significantly more stable, while the theoretical schedule is intentionally conservative, explaining the gap observed in DRL. We clarified this distinction in the revised version.
>
> ---
> **"Can you reduce the number of assumptions in the theoretical analysis?"**
>
> The reviewer is correct that the bounded-gradient assumption is stronger than strictly necessary. In our analysis it is used only to ensure that second-moment quantities (e.g., $E[g_k^2]$) remain finite. As in prior work on adaptive methods, this condition can be replaced by a weaker assumption such as a bounded second moment of the stochastic gradients ($E[‖g_k‖^2] < C$), without affecting the convergence rate or any qualitative conclusion.
>
> ---
>
> **"In terms of experiments, while results span CV/NLP/RL, the largest experiments remain moderate scale (e.g., CIFAR-level vision, GLUE-level language), so claims about large-scale pretraining or very long-horizon stability are untested, acknowledged in the Limitations. Do you have results beyond CIFAR-level vision and GLUE-level language, e.g., large-scale pretraining or very long-horizon runs, to support the broader claims?"}**
>
> Thank you for the question. We do not claim advantages for Ano in large-scale CV or NLP pretraining. Our empirical scope is intentionally aligned with the regime for which Ano was designed—high-noise and non-stationary settings such as RL. To make this more explicit, we added a Gradient Noise Scale analysis (Appendix B.3), which shows that Ano provides the largest gains when the stochasticity is high.
>
> The CIFAR-100 and GLUE experiments were included only as sanity checks to verify that Ano behaves reasonably outside its target regime. We clarified this in the revised version to avoid suggesting broader claims. Extending Ano to large-scale pretraining falls outside the present scope.
>
> ---
> **References :**
>
> [1] Bernstein, J., Wang, Y.-X., Azizzadenesheli, K., Anandkumar, A. (2018). signSGD: Compressed Optimization for Non-Convex Problems. ICML.
>
> [2] Dong, J., Li, H., Lin, Z. (2024). Convergence Rate Analysis of Lion. arXiv:2411.07724.
>
> [3] McCandlish, S., Kaplan, J., Amodei, D., et al. (2018). An Empirical Model of Large-Batch Training. OpenAI Technical Report.

---

### Meta-Review · Area_Chair_XXLg · 2026-01-07

**Summary:**

This paper introduces *Ano*, a first-order stochastic optimizer designed for noisy and non-stationary optimization with emphasis on RL. The core idea is to decouple the update direction and the magnitude, with the update direction determined by the momentum term, the step size scaling with the gradient and a Yogi-style second-moment term. A variant, *Anolog*, is also introduced, which removes sensitivity to the momentum hyperparameter by growing the momentum window according to a logarithmic schedule.

The authors provide non-convex convergence guarantees under standard smoothness and bounded-variance assumptions, establishing a rate of $O(T^{-1/4})$, which the authors claim is consistent with known results for sign-based methods.

Empirically, Ano is evaluated across computer vision (CIFAR), NLP (GLUE), and RL (MuJoCo, Atari) tasks. Results suggest that Ano is competitive in low-noise supervised settings and provides gains in noisy or non-stationary regimes, especially RL, where it can learn faster and achieve better loss. Ablations and diagnostics isolate the effects of direction-magnitude decoupling, further supporting the conclusions of improved performance in noisy regimes.

**Reviewer Concerns:**

Reviewer concerns were as follows:
1. *Theory (raised primarily by bWeQ, Motx, partially by BgzE and Wdzh)*: Several reviewers questioned whether the theoretical contribution meaningfully supports the paper’s claims. Concerns were raised by BgzE and Motx about the restrictiveness and redundancy of assumptions, particularly the simultaneous use of smoothness and bounded gradients, and about minor inconsistencies or omissions in the mathematical presentation (e.g., unspecified variables, denominator mismatches). The authors acknowledge that some assumptions are stronger than necessary and state that bounded gradients can be replaced by weaker moment conditions without changing results. They also correct mathematical inconsistencies, clarify algorithmic details, and rerun experiments affected by implementation errors. These issues are largely addressed. The strongest criticism (bWeQ) is that the convergence rate is strictly worse than SGD, and that the paper initially appeared to imply deeper theoretical justification for Ano’s empirical robustness. There was also concern that the smooth non-convex framework poorly models modern deep learning, and that sign-based methods may not be as fundamentally limited as claimed. The authors respond by reframing the role of theory, explicitly stating that it is not intended to motivate Ano nor to claim superiority over SGD, but rather to show that Ano is not worse than existing sign-based methods under standard assumptions. The particular response to bWeQ was excellent in my view, and it is a shame that they could not respond to it. My understanding is that the theory is now better-positioned and less overstated, but still remains largely disconnected from the main empirical motivation, leaving a lingering theory-practice gap.

2. *Empirical scope and scale of experiments (raised by Motx, 6kFo, Wdzh, BgzE)*: Multiple reviewers expressed concern that experiments are limited to relatively small-scale supervised tasks (CIFAR-100, GLUE) and that claims about robustness or long-horizon stability are not supported by large-scale CV/NLP or very long training runs. Some reviewers explicitly requested ImageNet, ViT, or LLM-scale experiments. The authors respond consistently that Ano is not designed for large-scale, low-noise, stationary regimes, and that the supervised experiments are included only as sanity checks. They reinforce this by adding Gradient Noise Scale analyses and clarifying the intended scope in the paper and limitations section. This response may not fully satisfy reviewers who view robustness claims as implicitly broader. That said, given the RL focus, the refusal to pursue large-scale pretraining experiments is defensible.

3. *Strength and interpretation of empirical gains (raised by bWeQ, Wdzh, 6kFo)*: Several reviewers noted that final performance often overlaps with baselines within confidence intervals, raising doubts about the significance of gains. Others pointed out higher variance in Ano’s results and questioned whether improvements are systematic. The authors assert that RL evaluation is high-variance and that overlapping CIs do not imply equal performance. They introduce aggregate metrics (IQM, normalized scores) and increase transparency around evaluation protocols. They also claim that higher variance reflects heavier upper tails rather than instability. While this response is reasonable, it is not entirely convincing. Some key aggregate statistics were added late, and the interpretation still relies heavily on qualitative judgment.

4. *Comparison with closely related methods (especially Grams) (raised by Wdzh)*: A notable concern was that the paper does not sufficiently explain why Ano outperforms Grams, despite their symmetric design choices. The reviewer requested deeper analysis, especially given anomalous behavior under low noise. The authors provide a plausible mechanistic explanation (momentum-based magnitude causing overdamping under noise) and add additional synthetic experiments. This is one of the more successful rebuttals, as it directly engages with the algorithmic distinction and improves interpretability.

5. *Hyperparameter sensitivity, robustness diagnostics, and practical details (raised by 6kFo, Motx)*: Concerns included sensitivity to batch size and replay buffer size, interaction with normalization layers, artificiality of noise injection, and inconsistent bias correction in code. The authors respond thoroughly by adding hyperparameter sensitivity studies, clarifying compatibility with BN/LN, justifying noise injection via GNS, and correcting implementation inconsistencies with rerun experiments. These responses are concrete and credible, and they materially improve the paper’s reliability.

**Reviewer Scores:**

While I would like to have seen responses from Reviewer bWeQ to the authors’ excellent responses there, I am skeptical that all negative reviewers would have sufficiently raised their scores during the discussion period for the paper to be accepted. There are quite a few modifications to the document, perhaps more than the reviewers would have comfortably signed off on. Several issues raised, particularly those involving the theory-practice gap, are unlikely to have been addressed during the discussion period. Reviewers gave few indications of willingness to significantly raise their scores.

---

### Decision · Program_Chairs · 2026-01-26

Reject